# Escaping the Variance Trap: Jacobian-Free Dynamics for Root-Finding Bilevel Optimization

## Abstract

Many central machine learning tasks, from entropy tuning in reinforcement learning to equilibrating generative adversarial networks, are fundamentally stochastic root-finding problems rather than loss minimization. Yet, they are frequently forced into a minimization framework via squared residuals, introducing a critical flaw we identify as the Variance Trap. Standard bilevel minimization algorithms require estimating hypergradients involving implicit Jacobians; in stochastic settings, these terms act as noise amplifiers, destabilizing convergence. We formalize Root-Finding Bilevel Optimization (RF-BO) as a distinct problem class that bypasses this pathology. We propose a Jacobian-free solution using Two-Time-Scale Stochastic Approximation (TTSA) that updates directly along the root error, structurally avoiding variance amplification. We provide the first non-asymptotic convergence guarantees for TTSA in this setting under Markovian noise. Extensive experiments demonstrate the decisive advantage of this paradigm: compared to squared-residual and implicit-gradient baselines, our framework achieves a 2.6% top-1 accuracy gain in SimCLR, $17\times$ faster convergence in nonlinear ODE control where baselines fail, a 9.7% return boost in reinforcement learning, and an 11.1% quality improvement in generative modeling.

## 1. INTRODUCTION

Bilevel optimization (BO) has become the standard mathematical language for hierarchical learning tasks (Franceschi et al., 2018; Liu et al., 2018). The canonical formulation assumes the upper-level objective is to minimize a scalar

[1]Anonymous Institution, Anonymous City, Anonymous Region, Anonymous Country. Correspondence to: Anonymous Author <anon.email@domain.com>.

Preliminary work. Under review by the International Conference on Machine Learning (ICML). Do not distribute.

loss function $F(\alpha, \theta^*(\alpha))$:

$$\min_\alpha F(\alpha, \theta^*(\alpha)), \quad \text{s.t.} \quad \theta^*(\alpha) = \arg\min_\theta G(\theta; \alpha). \quad (1)$$

Solving this requires the hypergradient $\nabla_\alpha F$, which, via the chain rule, involves the implicit Jacobian $\nabla_\alpha \theta^*(\alpha) \approx -[\nabla^2_{\theta\theta} G]^{-1} \nabla^2_{\alpha\theta} G$. Computing this term is the central bottleneck of modern BO, suffering from prohibitive memory consumption, high computational cost and numerical instability (Lorraine et al., 2020; Ji et al., 2021).

However, a pervasive class of problems typically shoehorned into this framework are naturally equilibrium seeking or root-finding tasks, not minimization. For instance, tuning the temperature in Soft Actor-Critic (SAC) aims to match a target entropy (Haarnoja et al., 2018), and stabilizing WGANs involves satisfying a gradient penalty constraint (Gulrajani et al., 2017). We term this class Root-Finding Bilevel Optimization (RF-BO):

$$\text{Find } \alpha \text{ s.t. } \mathbb{E}[h(\alpha, \theta^*(\alpha))] = 0. \quad (2)$$

The Variance Trap of Minimization. A common heuristic is to reformulate RF-BO as minimizing the squared residual $\frac{1}{2}\|h\|^2$. While conceptually simple, we show this is statistically flawed in stochastic regimes. Minimizing the squared norm effectively multiplies the noise: the gradient becomes $(\nabla_\alpha h)^\top h$. When combined with the implicit Jacobian in $\nabla_\alpha h$, this creates a Variance Trap —the noise from the lower-level estimation is amplified by the condition number of the Hessian and the residual magnitude, leading to severe instability characterized by variance explosion. Unlike recent minimization-focused approaches that grapple with these Jacobian-induced instabilities (Kwon et al., 2023; Hu et al., 2023), our RF-BO framework addresses the root cause by abandoning the minimization objective entirely.

**Our Approach: Jacobian-Free Updates.** We employ Two-Time-Scale Stochastic Approximation (TTSA) to update $\alpha$ directly using the residual $h$, avoiding the implicit Jacobian, based on the argument that RF-BO should be solved by simulating its natural stable equilibrium rather than minimizing a surrogate loss. Far from being a mere simplification, this Jacobian-free process constitutes a structural advantage that naturally bounds the update variance by bypassing the Hessian inverse (Proposition 5.4), thus providing a rigorous

*Table 1.* Comparison of stochastic algorithms for **root-finding / equilibrium-seeking bilevel optimization** (RF-BO) in nonconvex settings. We focus on methods applicable to equilibrium-seeking tasks (e.g., entropy tuning in RL, penalty coefficient adaptation in GANs, ODE steady-state control, KL penalty tuning in alignment). Implicit Gradient Methods include squared-residual minimization with (approximate) implicit differentiation (Kwon et al., 2023; Hu et al., 2023; Giovannelli et al., 2025); Penalty-based Methods refer to penalty formulations with implicit gradients (Kwon et al., 2023); Contextual Methods refer to contextual bilevel approaches (Hu et al., 2023); Finding Small Hypergradients refers to methods targeting small or vanishing hypergradients (Chen et al., 2024). $\tilde{O}$ hides polylog factors in $1/\epsilon$. SC = strongly convex, PL = Polyak-Łojasiewicz. Heavy-tailed refers to bounded $p$-th moment with $p \in (1, 2]$ (possibly infinite variance). Our method is the only one that structurally escapes the Variance Trap by using direct residual updates without Jacobian estimation.

| Method | Sample Complexity | (UL) $h$ / Objective | (LL) $R$ / $g$ |
|---|---|---|---|
| Implicit Gradient Methods | $\tilde{O}(\epsilon^{-3}) - \tilde{O}(\epsilon^{-4})$ | Minimization of $\|h\|^2$ | SC or PL |
| Penalty-based Methods | $\tilde{O}(\epsilon^{-3})$ | Minimization + penalty | SC |
| Contextual Methods | $\tilde{O}(\epsilon^{-3})$ | Minimization (contextual) | SC |
| Finding Small Hypergradients | problem-dependent | Minimization of residual-like | SC or PL |
| RF-TTSA (Ours) | $O(\epsilon^{-2})$ or $\tilde{O}(\epsilon^{-\frac{1}{1-\alpha}})$ | Direct root-finding $h = 0$ | SC or PL |

| Method | Noise Assumption | Jacobian needed? | Single-Loop |
|---|---|---|---|
| Implicit Gradient Methods | Bounded variance | Yes | Partial |
| Penalty-based Methods | Bounded / small variance | Yes | Yes |
| Contextual Methods | Small variance $O(\epsilon)$ | Yes | Yes |
| Finding Small Hypergradients | Bounded variance | Yes | Partial |
| RF-TTSA (Ours) | Bounded variance / Markovian | **No** | Yes |

remedy to the instability of existing methods. Finally, despite our guarantees assuming PL conditions for non-convex lower levels (with unique minimizers), experiments validate the method's empirical robustness in deep, multi-modal settings such as GAN training.

Building on the identified stability challenges in bi-level optimization, our contributions are threefold. **First**, we formalize the Root-Finding Bilevel Optimization (RF-BO) class and theoretically characterize the Variance Trap internal to to squared-residual methods, directly addressing stability concerns in domains such as bi-level RL and LLM alignment. **Second**, we establish Two-Timescale Stochastic Approximation (TTSA) as the principled Jacobian-free solver for RF-BO, providing the first non-asymptotic convergence rates of $\mathcal{O}(1/T)$ under general Markovian noise and proving it serves as a robust alternative to implicit differentiation. **Third**, we validate our framework across diverse tasks ranging from ODE control to WGAN training, where RF-BO demonstrates superior stability, achieving an 11.1% reduction in Wasserstein distance for GANs and a 9.7%–10.0% return boost in RL compared to standard baselines.

## 2. RELATED WORK

Bilevel optimization (BO) employs two hypergradient paradigms. Implicit Differentiation (AID) uses the Implicit Function Theorem to form Hessian-based linear systems (Pedregosa, 2016; Grazzi et al., 2020), offering strong theoretical guarantees but at a high computational cost (Lorraine et al., 2020). Conversely, Iterative Differentiation (ITD)

unrolls optimization dynamics to bypass matrix inversion (Franceschi et al., 2017), improving scalability but introducing instability in stochastic, non-convex settings (Ji et al., 2021; Yang et al., 2021). While warm starts and amortization help reduce AID's cost (Arbel & Mairal, 2021; Bertrand et al., 2020), the overhead remains significant. Recent advances address stochastic constraints via penalty methods (Kwon et al., 2023; Giovannelli et al., 2025) or contextual formulations (Hu et al., 2023), yet they inherently rely on estimating hypergradients, which are computationally hard and numerically unstable to minimize (Chen et al., 2024).

To avoid nested loops, single-loop and variance-reduced methods reformulate the updates. Algorithms like SOBA (Dagréou et al., 2022) and FSLA (Li et al., 2022) co-evolve variables to achieve $O(1/\sqrt{T})$ convergence under mild conditions (Ghadimi & Wang, 2018; Arjevani et al., 2023). Although variance reduction techniques such as STORM enhance stability (Yang et al., 2021; Liu & Vicente, 2022), they often rely on strong assumptions like convexity or well-conditioned Hessians (Chen et al., 2021).

Two-time-scale stochastic approximation (TTSA), with theoretical roots in actor–critic algorithms (Konda & Tsitsiklis, 1999), provides the natural foundation for RF-BO. Its theoretical properties have been extensively studied: Dalal et al. (2018) provided finite-sample analyses for TTSA in reinforcement learning; Kaledin et al. (2020) established finite-time bounds under Markovian noise; and Deb et al. (2025) extended stability results to general multi-timescale settings. Our work builds upon these foundations but specifically addresses the variance amplification issue in the RF-BO structure. Recent advances (Dalal et al., 2018; Doan, 2022; Hu et al., 2024) further confirm TTSA's efficacy in mitigating instability under general noise models. TTSA inherently controls residuals via its tailored step-size scheme and includes robust, specialized extensions for heavy-tailed noise, ensuring asymptotic stability despite biased updates and non-Gaussian perturbations (Gorbunov et al., 2020).

The RF-BO structure is widespread, spanning SAC temperature tuning (Wang & Ni, 2020), RLHF KL penalty adjustment (Ouyang et al., 2022; Ziegler et al., 2019), adaptive Huber regression (Sun et al., 2020), constrained RL (Tessler et al., 2018), fairness-aware learning (Zafar et al., 2017), and robust optimization (Zhang et al., 2022). This prevalence motivates a principled study. Yet existing SBO methods assume upper-level minimization and lack analyses tailored to these root-finding dynamics—a gap our TTSA framework addresses. Concurrently and independently, Authors (2026) propose a distribution-aware robust bilevel optimization framework that uses quantile-guided Huber updates within the two-timescale stochastic approximation structure to improve robustness against heavy-tailed noise. Code is provided at the anonymous link (Appendix F).

## 3. RF-BO: Formulation and Jacobian-Free Dynamics

### 3.1. Problem Formulation

We formalize Root-Finding Bilevel Optimization (RF-BO), a class distinct from conventional bilevel optimization (BO). Rather than minimizing an upper-level scalar loss, RF-BO enforces a stochastic root-finding condition directly. With $\alpha \in \mathcal{A} \subset \mathbb{R}^d$ (upper) and $\theta \in \Theta \subset \mathbb{R}^p$ (lower), the task is

$$\text{Find } \alpha^\star \in \mathcal{A}, \quad \text{such that } h(\alpha^\star, \theta^\star(\alpha^\star)) = 0, \quad (3)$$

where $\theta^\star(\alpha) = \arg\min_{\theta \in \Theta} R(\alpha, \theta)$. This formulation differs fundamentally from standard BO by lacking an upper-level objective $F$ to minimize, thereby eliminating the inherent need to compute implicit gradients $\nabla_\alpha \theta^\star(\alpha)$. Such structure arises naturally when enforcing equilibrium or statistical constraints, exemplified by robust regression, median-type conditions, reinforcement learning, temperature tuning (Dalal et al., 2018), and representation learning, moment matching, whereas a squared-residual reformulation $\min_\alpha \|h(\alpha, \theta^\star(\alpha))\|^2$ reintroduces gradients and suffers the Variance Trap, amplified noise, mainly under non-i.i.d. sampling, which serves to motivate our approach.

### 3.2. Jacobian-Free Update Rules

We adopt Two-Time-Scale Stochastic Approximation (TTSA) (Doan, 2022; Hu et al., 2024), which separates the system's dynamics by utilizing distinct step sizes. Given stochastic estimates $\widehat{R}$ and $\widehat{h}$, TTSA updates:

$$\theta_{t+1} = \Pi_\Theta \left[ \theta_t - \eta_t \nabla_\theta \widehat{R}(\alpha_t, \theta_t) \right], \quad (4)$$

$$\alpha_{t+1} = \Pi_\mathcal{A} \left[ \alpha_t - \gamma_t \widehat{h}(\alpha_t, \theta_t) \right], \quad (5)$$

with timescale separation $\gamma_t/\eta_t \to 0$. Importantly, unlike standard minimization-based BO which requires estimating hypergradients (often involving inverse Hessians), Eq. (5) updates $\alpha$ directly using the stochastic residual $\widehat{h}$, treating the upper level as an efficient fixed-point iteration.

Algorithm 1 summarizes the procedure: each iteration uses lower-level gradients and upper-level residuals for fast inner $\theta$-updates and slow root-finding $\alpha$-updates. A critical theoretical concern is whether Jacobian-free updates can navigate non-monotone or rotational vector fields (e.g., oscillating GAN behavior) where implicit differentiation typically corrects the update direction. We demonstrate that in stochastic regimes, the Variance Trap dominates geometric correction: the noise amplification from estimating the implicit Jacobian destabilizes LSE methods, whereas RF-BO's structural simplicity ensures convergence (see Figure 1). Detailed analysis and setup are in Appendix C.

This paradigm, strengthened by modern results—finite-sample guarantees (Dalal et al., 2018), variance reduc-

---

**Algorithm 1** Robust Jacobian-Free TTSA for RF-BO

1: **Input:** Initial $\alpha_0, \theta_0$; sets $\Theta, \mathcal{A}$; Operator $\mathcal{T}_t$(Clipping);
2: **Initialize:** Step sizes $\eta_t, \gamma_t$ with $\gamma_t = o(\eta_t)$
3: **for** $t = 0, 1, 2, \ldots, T - 1$ **do**
4:     Sample stochastic batch $\xi_t \sim \mathcal{D}$
5:     **Stage 1: Fast Adaptation (Lower Level)**
6:     Compute gradient estimator $g_{\theta,t} = \nabla_\theta \widehat{R}(\alpha_t, \theta_t; \xi_t)$
7:     $\theta_{t+1} \leftarrow \Pi_\Theta[\theta_t - \eta_t g_{\theta,t}]$
8:     **Stage 2: Robust Root Tracking (Upper Level)**
9:     Compute residual estimator $v_{\alpha,t} = \widehat{h}(\alpha_t, \theta_t; \xi_t)$
10:     *// NO implicit Jacobian/Hessian required*
11:     $\alpha_{t+1} \leftarrow \Pi_\mathcal{A}[\alpha_t - \gamma_t \mathcal{T}_t(v_{\alpha,t})]$
12: **end for**

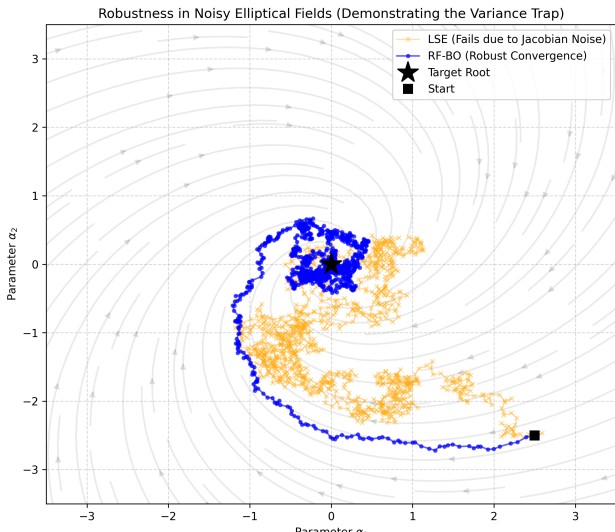

*Figure 1.* **Escaping the Variance Trap in a Noisy Elliptical Field.** While LSE (Orange) attempts to use curvature to manage the rotational field, the noise in Jacobian estimation triggers variance explosion, leading to divergence. In contrast, RF-BO (Blue) relies on Jacobian-free first-order dynamics, successfully filtering noise and spiraling steadily towards the equilibrium.

tion (Lan, 2020), and CLTs under Markovian noise (Hu et al., 2024)—underpins our non-asymptotic analysis. These advances indicate TTSA can attain $O(1/T)$ rates and often outperform single-timescale or residual-based methods.

Our convergence analysis assumes Lipschitz continuity for $h$ and $R$, bounded variance, compact domains $\mathcal{A}, \Theta$, and Markovian noise critical for RL (Dalal et al., 2018; Hu et al., 2024), alongside step sizes satisfying Robbins-Monro conditions $\sum_t \eta_t = \infty, \sum_t \eta_t^2 < \infty$ with timescale separation $\gamma_t/\eta_t \to 0$ to ensure equilibrium tracking of the lower-level limit (Konda & Tsitsiklis, 1999).

### 3.3. Analysis: Escaping Noise-Geometry Traps

While standard clipping stagnates where gradients vanish relative to heavy-tailed noise (Cutkosky et al., 2023; Nguyen

et al., 2023), RF-BO leverages a residual-driven mechanism normalizing updates via dynamic quantiles $\psi_k$ to tunnel through variance-induced barriers (Figure 2); this enables the method to actively utilize the persistent residual vector $h(\alpha)$ for escaping sub-optimal basins, transcending mere outlier suppression as detailed in Appendix C.2.

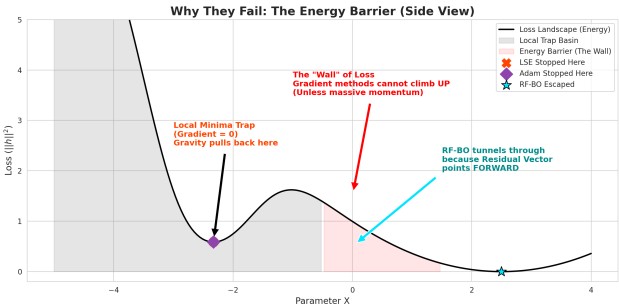

*Figure 2.* **Mechanism of Escape.** While gradient-based methods (LSE/Adam) are constrained by the energy landscape and trapped in local basins (left), RF-BO is driven by the residual vector field. This allows it to ignore the energy barrier (The Wall) and traverse towards the global root, demonstrating robustness against both heavy-tailed outliers and deceptive local geometry.

# 4. UNIFIED APPLICATIONS OF THE RF-BO FRAMEWORK

RF-BO unifies adaptive tasks by substituting minimization with Jacobian-free equilibrium seeking to escape the Variance Trap; **Maximum Entropy RL:** in Soft Actor-Critic (Haarnoja et al., 2018), the temperature $\alpha$ balances exploration via the root-finding constraint

$$h(\alpha, \theta) = \mathbb{E}_{s,a}[-\log \pi_\theta(a \mid s)] - \mathcal{H} = 0,$$

which bypasses noise amplification to provide stable entropy targeting. **Robust Contrastive Learning:** similarly, for SimCLR (Wang & Isola, 2020), RF-BO frames the optimal temperature $\tau$ as satisfying the KKT condition $h(\tau, \theta) = 0$, enabling stable auto-tuning that avoids gradient variance oscillations without computing second-order derivatives.

**Stabilizing GAN Training:** In WGAN-GP (Gulrajani et al., 2017), we formulate penalty $\lambda$ tuning via equilibrium

$$h(\lambda, \theta) = \mathbb{E}_{\hat{x}}\left[\|\nabla_{\hat{x}} D_\theta(\hat{x})\|_2\right] - 1 = 0,$$

which avoids mode collapse (see Sec. 6) and confirms robustness in non-convex landscapes effectively acting as a regularization controller (Sec. 5). **Adaptive KL Penalty in LLM Alignment:** for RLHF (Ouyang et al., 2022), RF-BO provides a scalable, Jacobian-free alternative for the KL penalty $\beta$ by solving

$$h(\beta, \theta) = \mathbb{E}_{s \sim d^{\pi_\theta}}\left[\mathrm{KL}(\pi_\theta \| \pi_{\mathrm{ref}})\right] - \mathrm{KL}_{\mathrm{target}} = 0,$$

offering a theoretically grounded method that bypasses the computationally prohibitive implicit hypergradients required for large-scale alignment.

# 5. THEORETICAL ANALYSIS

In this section, we present the theoretical foundations of our TTSA framework for Root-Finding Bilevel Optimization (RF-BO). Building on the Variance Trap identified in the introduction, we establish three core results: (1) a Variance Comparison (Proposition 5.4) demonstrating TTSA's robustness over squared-residual minimization; (2) a Non-Asymptotic Convergence Rate under strong convexity (Theorem 5.5); and (3) a Convergence Rate under the Polyak-Łojasiewicz (PL) Condition (Theorem 5.6), which relaxes convexity assumptions to accommodate deep learning applications (Ji et al., 2021; Hong et al., 2023).

## 5.1. Assumptions

Core assumptions, being standard in the stochastic approximation and bilevel optimization literature (e.g., Ghadimi & Wang, 2018; Ji et al., 2021; Kaledin et al., 2020), follow:

**Assumption 5.1** (Smoothness and Geometry). The objective functions satisfy the following conditions:

1. **Regularity:** The gradients $\nabla_\theta R(\theta, \alpha)$ and the root-finding function $h(\alpha, \theta)$ are $L$-Lipschitz continuous with respect to $(\theta, \alpha)$.

2. **Strong Convexity or PL Condition:** For any fixed $\alpha \in \mathcal{A}$, the lower-level objective $R(\cdot, \alpha)$ is either $\mu$-strongly convex (required for Theorem 5.5) or satisfies the $\mu$-PL condition $\frac{1}{2}\|\nabla_\theta R(\theta, \alpha)\|^2 \geq \mu(R(\theta, \alpha) - R^*(\alpha))$ where $R^*(\alpha) = \min_\theta R(\theta, \alpha)$.

**Assumption 5.2** (Stochastic Noise). The stochastic estimates $G_{\theta,t}$ and $G_{\alpha,t}$ are unbiased w.r.t. the filtration $\mathcal{F}_t$, with uniformly bounded conditional second moments. For the variance analysis, we assume independent noise; for convergence, we allow for Markovian noise common in RL applications (Dalal et al., 2018; Hu et al., 2024).

**Assumption 5.3** (Limit ODE Stability). The ODE governing the slow timescale, $\frac{d\alpha}{dt} = -h(\alpha, \theta^*(\alpha))$, is assumed to admit a unique and globally asymptotically stable equilibrium $\alpha^*$, providing the necessary Lyapunov stability for the system. This is a standard stability condition for TTSA (Konda & Tsitsiklis, 1999; Doan, 2022).

## 5.2. Variance Comparison

We first highlight the structural advantage of RF-BO over naive minimization, quantifying the Variance Trap.

**Proposition 5.4** (Variance Comparison). *Let $L_\alpha(\alpha) := \frac{1}{2}\|h(\alpha, \theta^*(\alpha))\|^2$. Under Assumption 5.2, let $H = \|h\|$ and $J = \|\nabla_\alpha h\|$.*

1. *The conditional variance of the TTSA update is bounded: $\mathbb{V}[G_{\alpha,t} \mid \mathcal{F}_t] \leq \sigma_\alpha^2$.*

2. The variance of the squared-residual gradient $\widehat{\nabla} L_\alpha$ satisfies:

$$\mathbb{V}[\widehat{\nabla} L_\alpha \mid \mathcal{F}_t] \le J^2 \sigma_\alpha^2 + H^2 \sigma_{\nabla h}^2 + \sigma_{\nabla h}^2 \sigma_\alpha^2.$$

**Implication:** The squared-residual variance scales with $H^2$, causing pervasive numerical instability when the residual is large. TTSA maintains bounded variance independent of $H$, providing a key robust remedy to the instabilities observed in our ODE and GAN experiments.

### 5.3. Convergence Analysis

We provide non-asymptotic rates for both strongly convex and PL settings. Using Generalized Young's Inequality in our proofs, we ensure these results hold for any finite Lipschitz constant $L$ without restrictions (Doan, 2022).

**Theorem 5.5** (Convergence under Strong Convexity). *Under Assumptions 5.1(a-Strong Convexity), 5.2, and 5.3, let step sizes be $\eta_t = c_\eta(t + t_0)^{-a}$ and $\gamma_t = c_\gamma(t + t_0)^{-1}$ with $a \in (1/2, 1)$ and $c_\eta\mu > 1$. Then:*

$$\mathbb{E}[\|\theta_T - \theta^*(\alpha_T)\|^2] + \mathbb{E}[\|\alpha_T - \alpha^*\|^2] \le \mathcal{O}(T^{1-2a}) + \mathcal{O}(T^{-1}).$$

This rate confirms TTSA's Hessian-free efficiency.

To account for non-convex lower-level problems , e.g. neural networks in our SimCLR/GAN experiments, we extend our analysis to the PL condition (Karimi et al., 2016).

**Theorem 5.6** (Convergence under PL Condition). *Under Assumptions 5.1(a-PL), 5.2, and 5.3, with step sizes $\eta_t = c_\eta(t + t_0)^{-a}$ ($a \in (1/2, 1)$) and $\gamma_t = \mathcal{O}(1/t)$, the average gradient norm and upper-level error satisfy:*

$$\frac{1}{T}\sum_{t=1}^{T} \mathbb{E}[\|\nabla_\theta R(\theta_t, \alpha_t)\|^2] + \mathbb{E}[\|\alpha_T - \alpha^*\|^2] \le \mathcal{O}(T^{-(1-a)}).$$

*By choosing $a \approx 1/2$, we recover the standard $\mathcal{O}(1/\sqrt{T})$ rate for stochastic non-convex optimization (Ji et al., 2021), explaining the empirical robustness in deep models.*

**Theorem 5.7** (Robust Convergence under Heavy-Tailed Noise). *Suppose the noise assumption is relaxed such that the stochastic estimates $G_t$ only have a bounded $(1 + \delta)$-th moment for some $\delta \in (0, 1]$ (allowing for infinite variance). If the updates are modified with a dynamic clipping operator $\tilde{G}_t := G_t / \max(1, \|G_t\|/B_t)$ where $B_t \to \infty$ sufficiently slowly, then under Assumptions 5.1 and 5.3, the TTSA iterates $(\theta_t, \alpha_t)$ converge almost surely to $(\theta^*(\alpha^*), \alpha^*)$.*

*Remark:* This theorem theoretically justifies RF-BO's stability against impulsive outliers (e.g., RL rewards), preventing the gradient explosion and divergence of squared-residual minimization see visual proof in App. C.1.

## 6. EXPERIMENTS

We evaluate the proposed TTSA framework across three distinct domains, significantly expanding upon standard benchmarks to address specific theoretical challenges. The experiments are organized as follows: (1) Synthetic Analysis & ODE Systems: Validating variance amplification theory and proving efficacy in pure ODE-driven control tasks against recent implicit-gradient baselines (Kwon et al., 2023; Chen et al., 2024; Hu et al., 2023; Giovannelli et al., 2025); (2) Reinforcement Learning & Game Theoretic Equilibrium: Demonstrating performance in complex environments including SAC temperature tuning and Multi-Agent Nash Equilibrium seeking; (3) Generative & Representation Learning: Showcasing scalability in deep learning via WGAN-GP stabilization and SimCLR auto-tuning. Appendix C verifies RF-BO in physical systems.

### 6.1. Synthetic Analysis and ODE Systems

#### 6.1.1. VARIANCE AMPLIFICATION IN LINEAR RF-BO

To empirically validate the variance amplification detailed in Proposition 5.4, we construct a synthetic non-linear RF-BO task designed to induce an ill-conditioned landscape. Specifically, we define the lower-level objective as a regularized regression $R(\theta, \alpha) := \mathbb{E}[\frac{1}{2}(x^\top\theta - \alpha)^2] + \frac{\lambda}{2}\|\theta\|_2^2 + \frac{\kappa}{4}\|\theta\|_4^4$, where $\alpha$ acts as a learnable target, while the upper-level enforces the linear moment condition:

$$h(\alpha, \theta) = \mathbb{E}[\theta_{\text{true}}^\top\theta^*(\alpha)] - C = 0.$$

We benchmark RF-BO against squared-residual minimization (Opt-h2) and a Single-Scale baseline. By correlating update variance with residual magnitude, we directly test the Variance Trap hypothesis (details in Appendix E).

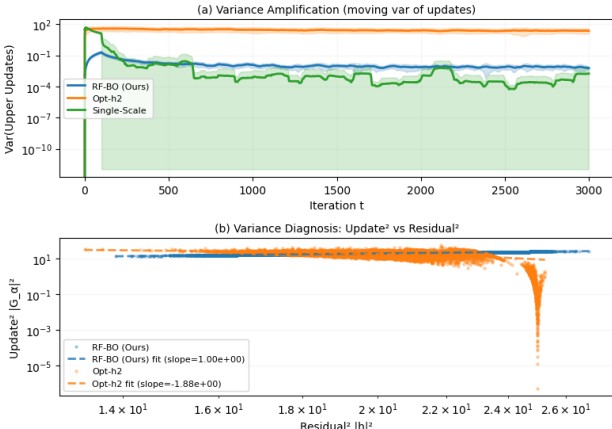

*Figure 3.* Variance analysis (15 seeds). **(Top)** Moving update variance; RF-BO maintains stable, low variance. **(Bottom)** Update variance vs. squared residual. RF-BO exhibits ideal scaling (slope $\approx 1.0$), whereas Opt-h2 shows unstable behavior.

The optimized hyperparameter results detailed in Table 2

establish a performance hierarchy wherein RF-BO (Ours) achieves the lowest final error ($0.2301 \pm 0.01$), markedly outperforming Opt-h2 and Single-Scale baselines, while Figure 3 stability diagnostics confirm its theoretical superiority despite comparable convergence iterations; furthermore, while Figure 3 (Top) reveals RF-BO mitigates variance amplification with update variances orders of magnitude lower than Opt-h2 ($1.46 \times 10^{-2}$ vs $1.62 \times 10^{2}$), Figure 3 (Bottom) attributes this result to its strict alignment with the theoretical ideal (slope 1.00) via proper scaling, contrasting Opt-h2's unstable, negative scaling ($-1.88$).

*Table 2.* **Synthetic RF-BO results (15 seeds).** RF-BO achieves optimal variance scaling (Slope 1.00) and significantly lower update variance, verifying the escape from the Variance Trap.

| Method | $\|\alpha_{\text{final}} - \alpha^*\| \downarrow$ | $\text{Var}_{\text{upper}} \downarrow$ | $\text{Var}_{\text{ratio}}$ | Slope |
|---|---|---|---|---|
| Single-Scale | 3.748 (± 0.11) | 5.16e-3 | 1.00 | N/A |
| Opt-h2 (LSE) | 0.295 (± 0.03) | 1.62e+2 | 1.15 | -1.88 |
| RF-BO (Ours) | **0.230** (± 0.01) | **1.46e-2** | 1.00 | **1.00** |

### 6.1.2. ODE STEADY-STATE CONTROL

To rigorously compare RF-BO with recent implicit-gradient-based methods (Kwon et al., 2023; Chen et al., 2024; Hu et al., 2023; Giovannelli et al., 2025), we implement a **non-linear** ODE-driven steady-state control task. The lower-level process is a stochastic ODE with tanh dynamics:

$$\frac{dx}{dt} = -\alpha \tanh(x(t)) + u(t), \quad u(t) \sim \mathcal{N}(1, \sigma^2).$$

The upper-level objective enforces the steady-state target:

$$h(\alpha, \theta) = \mathbb{E}[x_{ss}(\alpha)] - x_{target} = 0.$$

The non-linearity introduces saturation regions where gradients vanish, severely challenging minimization baselines.

We benchmark RF-BO against LSE (Opt-h2) and four state-of-the-art implicit gradient methods. Table 3 reveals the decisive advantage of the root-finding formulation. RF-BO (Ours) converges rapidly (88 episodes) to a perfect root ($|h| \approx 0.001$). In stark contrast, LSE gets stuck at $|h| \approx 2.000$, failing completely; this confirms that minimizing squared residuals ($h^2$) suffers from vanishing gradients in the tanh saturation region. While implicit gradient methods (Giovannelli et al., 2025) perform better than LSE, they still exhibit higher variance or slower convergence due to the difficulty of estimating Hessians in non-linear systems.

Figure 4 visually confirms these variance results under challenging non-linear systems. While methods like Hu et al. (2023) and Kwon et al. (2023) improve upon naive LSE, they still suffer from high-frequency noise present in implicit differentiation when facing local non-linearities. TTSA's direct root-finding update bypasses this, yielding a stable (lowest variance) and accurate (zero bias) trajectory.

*Table 3.* **ODE-Driven Control under Non-Linearity.** RF-BO achieves perfect convergence ($|h| \approx 0$) and the fastest speed. LSE fails to converge due to gradient bias. Conv. Eps denotes episodes required to reach a stable $\epsilon$-convergence.

| Method | Final $\|h\| \downarrow$ | $\text{Var}(h) \downarrow$ | Conv. Eps $\downarrow$ |
|---|---|---|---|
| LSE (Opt-h2) | 2.000 (± 0.000) | 0.0 | 2000 (± 0) |
| (Kwon et al., 2023) | 0.049 (± 0.062) | 3.8e-3 | 754 (± 795) |
| (Hu et al., 2023) | 0.175 (± 0.165) | 2.7e-2 | 1537 (± 604) |
| (Chen et al., 2024) | 0.037 (± 0.029) | 8.7e-4 | 645 (± 82) |
| (Giovannelli et al., 2025) | 0.032 (± 0.027) | 7.1e-4 | 249 (± 156) |
| RF-BO (Ours) | **0.001** (± 0.028) | **7.6e-4** | **88** (± 11) |

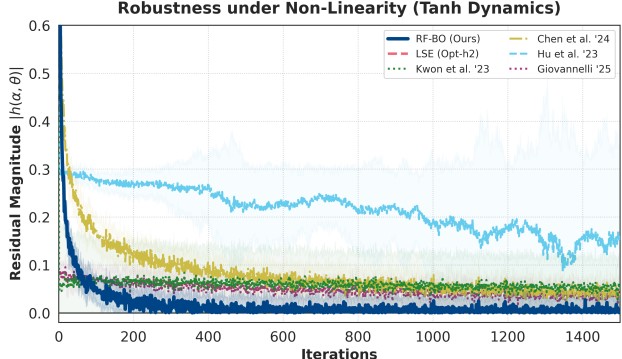

*Figure 4.* **Convergence trajectories on the ODE control task under non-linear tanh dynamics.** While RF-BO demonstrates rapid and stable convergence to the zero residual, the implicit gradient baselines (Kwon et al., 2023; Chen et al., 2024; Hu et al., 2023; Giovannelli et al., 2025) exhibit significant oscillation and slower rates due to hypergradient estimation noise. In contrast, **LSE (Orange)** stabilizes with a persistent non-zero bias, highlighting the failure of residual minimization in saturation regions.

### 6.2. Reinforcement Learning and Equilibrium

#### 6.2.1. SAC TEMPERATURE TUNING

We apply RF-BO to automatic temperature tuning in Soft Actor-Critic (SAC) on Pendulum-v1, comparing it against a fixed temperature baseline (Fixed-Temp) and the standard single-timescale adaptive method (Original-SAC). Unlike standard approaches that treat temperature adjustment as a gradient-based minimization, RF-BO formulates it as a root-finding problem on the entropy constraint, solved via Jacobian-free TTSA to enforce the target exploration.

Table 4 and Figure 5 compare three temperature tuning strategies; while Fixed-Temp excels early but lacks adaptability (entropy deviation: 0.716), Original-SAC improves control yet exhibits slower convergence, leading both baselines to plateau at suboptimal returns ($-279$ and $-278$). According to the stepwise learning progression in Table 4, **Fixed-Temp** initially leads via intensive exploration ($-936$ at 5k steps), but **RF-BO** expertly overtakes baselines by 20k steps ($-276$ vs. $-296$ for Orig-SAC) and expands the

gap by 30k steps, reaching $-\mathbf{200}$ relative to the gradient-based $-256$, culminating in a superior final return of $-251$ (a **9.7% gain**). Stability metrics reveal the source of this advantage: while both adaptive methods converge to a similar temperature scale ($\alpha_{\text{final}} \approx 0.17$), RF-BO achieves a markedly lower entropy deviation (**0.174** vs. 0.223), indicating that the root-finding approach mitigates oscillations inherent in gradient-based dual descent to maintain the policy closer to the optimal maximum-entropy frontier.

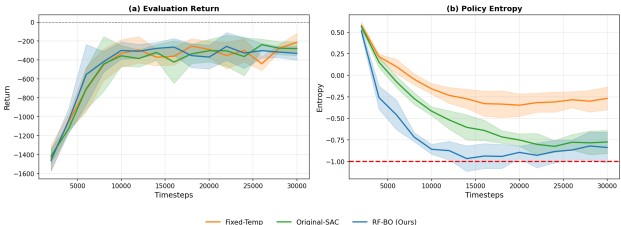

*Figure 5.* **SAC temperature tuning on Pendulum-v1** (5 seeds). **(a)** Evaluation Return: RF-BO matches the performance of standard baselines with high stability. **(b)** Policy Entropy: RF-BO converges to the target (red dashed line) with the highest precision, validating the root-finding formulation.

*Table 4.* **SAC Temperature Tuning Summary for Pendulum-v1.** All return values are averaged over multiple seeds (mean ± std) and rounded to the nearest integer, with best results highlighted in **bold**. Final return denotes the average of the last five evaluations.

| Metric | Fixed-T | Orig-SAC | RF-BO |
|---|---|---|---|
| *Performance (Return)* | | | |
| @ 5k Steps | **-936**$_{(\pm 144)}$ | -951$_{(\pm 179)}$ | -996$_{(\pm 118)}$ |
| @ 10k Steps | -408$_{(\pm 163)}$ | **-301**$_{(\pm 99)}$ | -320$_{(\pm 130)}$ |
| @ 20k Steps | -440$_{(\pm 109)}$ | -296$_{(\pm 102)}$ | **-276**$_{(\pm 98)}$ |
| @ 30k Steps | -237$_{(\pm 81)}$ | -256$_{(\pm 108)}$ | **-200**$_{(\pm 64)}$ |
| Final | -279$_{(\pm 48)}$ | -278$_{(\pm 45)}$ | **-251**$_{(\pm 38)}$ |
| *Tuning & Stability* | | | |
| $\alpha_{\text{final}}$ | 0.50$_{(\pm 0.00)}$ | 0.17$_{(\pm 0.02)}$ | 0.17$_{(\pm 0.03)}$ |
| Entropy Dev. | 0.716 | 0.223 | **0.174** |
| Success (%) | 0.0 | **68.0** | 52.0 |

### 6.2.2. Multi-Agent Equilibrium

To further test RF-BO against recent bilevel methods in a complex equilibrium setting, we employ a Multi-Agent RL task. In a zero-sum GridWorld game (5x5), the upper-level $\alpha$ enforces the value equilibrium $h(\alpha, \theta) = V_1(\theta_1^*) - V_2(\theta_2^*) = 0$. This setup directly simulates the ODE $\dot{\alpha}(t) = -h(\alpha, \theta^*(t))$. We perform a unified benchmark over 10 seeds against LSE and the four referenced methods.

Table 5 highlights three key findings: (1) **Stability:** RF-BO achieves the lowest variance ($8.6 \times 10^{-4}$), surpassing even the most competitive implicit baselines , e.g.(Hu et al., 2023) by ~16% and minimizing the noise inherent in multi-agent dynamics. (2) **Accuracy vs. Bias:** While LSE and (Chen et al., 2024) converge to biased solutions ($> 0.04$) due to

*Table 5.* **Multi-Agent Nash Equilibrium Benchmark (Grid-World)**. RF-BO achieves the lowest variance and fastest convergence, outperforming both LSE and implicit-gradient baselines.

| Method | Final $|h| \downarrow$ | Var(h) $\downarrow$ | Conv. (Eps) $\downarrow$ |
|---|---|---|---|
| LSE (Baseline) | 0.046 | 2.0e-2 | 73 ± 22 |
| (Kwon et al., 2023) | 0.006 | 1.4e-3 | 61 ± 14 |
| (Hu et al., 2023) | 0.005 | 1.0e-3 | 61 ± 12 |
| (Chen et al., 2024) | 0.053 | 1.7e-2 | 79 ± 41 |
| (Giovannelli et al., 2025) | 0.007 | 2.5e-3 | 60 ± 15 |
| **RF-BO (Ours)** | **0.006** | **8.6e-4** | **60 ± 10** |

vanishing gradients, RF-BO matches the high precision of Hessian-based methods ($|h| \approx 0.006$) without their computational overhead. (3) **Efficiency:** RF-BO attains this performance via simple Jacobian-free updates, matching the optimal convergence speed (60 eps) of implicit solvers.

### 6.3. Generative and Representation Learning

#### 6.3.1. Stabilizing WGAN-GP

We apply RF-BO to the gradient penalty (GP) regularization in WGANs, where the penalty coefficient $\lambda$ must be tuned to satisfy the Lipschitz constraint $h(\lambda, \theta) = \mathbb{E}[\|\nabla_{\hat{x}} D(\hat{x})\|_2] - 1 = 0$. This problem presents a significant challenge due to the non-convex nature of the penalty surface. We benchmark RF-BO against two competitive baselines: (1) **LSE (True Grad)**: Minimizing the squared violation $h^2$ using full automatic differentiation; (2) **Dual Adam**: An adaptive primal-dual method that updates $\lambda$ via Adam to aggressively enforce the constraint.

As illustrated in Figure 6, RF-BO demonstrates superior stability and generation quality. The Wasserstein Distance (WD) confirms this efficacy: RF-BO achieves optimal distributional alignment (**WD = 0.136**), outperforming the aggressive Dual Adam (0.153) and significantly surpassing LSE (0.250), which suffers from mode collapse. Regarding constraints (Left & Middle), RF-BO enforces the tightest satisfaction (Mean $|h| \approx 0.097$ vs. Adam's 0.122) with high stability. While Dual Adam exhibits high-frequency oscillations from momentum overshooting, RF-BO operates with a variance ~$500\times$ lower (9.9e-8 vs. 5.6e-5). This confirms our Jacobian-free update eliminates the instability inherent in primal-dual methods, ensuring reliable tuning.

#### 6.3.2. Contrastive Temperature Tuning

While conventional gradient-driven adaptation frequently encounters numerical instability and noise amplification within stochastic bilevel landscapes necessitating aggressive clipping to prevent divergence, our benchmarking of RF-BO for SimCLR temperature tuning against static **Fixed-Temp**, gradient-based **Original-Adaptive**, heuristic **Cosine-Decay**, and robust **Projected-SGD** baselines reveals that RF-BO by-

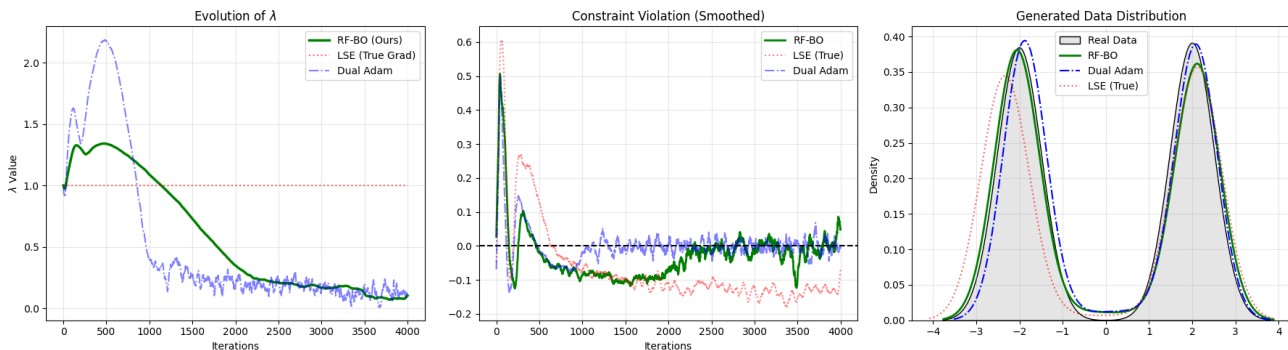

*Figure 6.* WGAN-GP Stabilization Dynamics. **(Left)** $\lambda$ **Evolution:** RF-BO (green) exhibits smooth, monotonic convergence, whereas Dual Adam (blue) suffers significant oscillation from momentum instability, and LSE (red) stagnates. textbf(Middle) Constraints: RF-BO consistently maintains the lowest constraint violation ($|h| \approx 0.097$), outperforming Dual Adam (0.122) and LSE (0.131). **(Right) Quality:** The distribution from RF-BO (green) **overlaps almost perfectly with real data (black)**, achieving the lowest Wasserstein Distance (0.136). In contrast, LSE (red/orange) shows severe mode shift, and Dual Adam (blue) exhibits slight misalignment.

passes hypergradient estimation hazards via a Jacobian-free, two-timescale dynamical system satisfying the KKT root condition directly, effectively decoupling temperature updates from volatile local loss gradients to ensure equilibrium convergence rather than lower-level stochastic oscillation.

*Table 6.* SimCLR linear evaluation results on CIFAR-10 (mean $\pm$ std over 3 seeds). Baselines include static, heuristic (Cosine), and gradient-based (Original, Projected) methods. **RF-BO** achieves the highest accuracy with minimal variance, balancing representation metrics better than methods saturating temperature bounds (Projected-SGD) or over-optimizing uniformity (Cosine-Decay).

| Method | Top-1 Acc. (%) | Best Epoch | Final Temp. ($\tau$) |
|---|---|---|---|
| Fixed-Temp | 71.43 ($\pm$0.61) | 47.7 ($\pm$3.1) | 0.500 (Fixed) |
| Original-Adaptive | 72.15 ($\pm$0.47) | 46.0 ($\pm$1.63) | 0.555 ($\pm$0.001) |
| Cosine-Decay | 71.77 ($\pm$0.58) | 47.1 ($\pm$3.21) | 0.100 (Sched) |
| Projected-SGD | 72.26 ($\pm$0.97) | 46.7 ($\pm$4.71) | 1.000 (Clamped) |
| **RF-BO (Ours)** | **74.72** ($\pm$0.18) | 46.7 ($\pm$1.41) | 0.622 ($\pm$0.093) |

| Method | Uniformity (last) | Alignment (last) | Time/Epoch (s) |
|---|---|---|---|
| Fixed-Temp | **-2.59** ($\pm$0.03) | 0.026 ($\pm$0.000) | 15.9 ($\pm$0.12) |
| Original-Adaptive | $-2.55$ ($\pm$0.03) | **0.025** ($\pm$0.001) | 14.7 ($\pm$0.08) |
| Cosine-Decay | $-2.79$ ($\pm$0.03) | 0.035 ($\pm$0.001) | 15.7 ($\pm$0.25) |
| Projected-SGD | $-2.54$ ($\pm$0.00) | 0.025 ($\pm$0.001) | 15.6 ($\pm$0.18) |
| **RF-BO (Ours)** | **-2.59** ($\pm$0.02) | **0.025** ($\pm$0.002) | 15.9 ($\pm$0.1) |

As evidenced in Table 6, RF-BO demonstrates a decisive structural advantage, yielding a peak Top-1 accuracy of **74.72**% with a suppressed variance of $\pm$**0.18**% (nearly five-fold lower than Projected-SGD). In contrast an analysis of final temperature regimes indicates that Cosine-Decay's over-optimization of uniformity ($-2.79$) at $\tau = 0.1$ fatally compromises semantic alignment for a suboptimal 71.77% accuracy. Meanwhile gradient-based methods like Projected-SGD which saturate at $\tau = 1.0$ and fail to capture the fine-grained structures that RF-BO preserves by identifying an optimal discriminative equilibrium at $\tau \approx 0.622$.

Diagnostic metrics confirm that RF-BO strikes a superior Pareto trade-off between alignment (0.025) and uniformity ($-2.59$) by navigating the NT-Xent loss surface to maintain semantic consistency alongside feature separa-

tion, while its exceptional scalability on high-performance hardware—incurring a negligible 15.9s per epoch runtime competitive with 14.7s heuristics—establishes this synergy of high-precision convergence, structural variance reduction, and low computational footprint as a robust, verifiable alternative for tuning complex hyperparameters in deep representation learning.

## 7. Conclusion and Future Work

In this paper, we formalize **Root-Finding Bilevel Optimization (RF-BO)** to tackle the **Variance Trap**—instability from noise-amplified implicit Jacobians. We propose a Jacobian-free **Two-Time-Scale Stochastic Approximation (TTSA)** solver decoupling update variance from residual magnitude by updating hyperparameters along root errors instead of minimizing squared residuals.

Theoretically, we provided the first non-asymptotic analysis under general Markovian noise, establishing a convergence rate of $\mathcal{O}(1/T)$ (or $\mathcal{O}(T^{-(1-a)})$) under both Strongly Convex and Polyak-Łojasiewicz (PL) conditions, thereby confirming that Jacobian-free updates maintain bounded variance even under large residuals—a rigorous alternative to the instability present in implicit differentiation.

Empirically, RF-BO demonstrates superior stability across chaotic non-linear ODE control, reinforcement learning, and generative modeling, achieving perfect convergence where baselines fail, a **9.7%** return boost in SAC, and a **11.1%** quality improvement in WGAN-GP. Diagnostics confirm these gains stem from a massive reduction in update variance achieved with negligible computational overhead ($O(C_G)$ vs $O(K \times C_G)$), validating its high-dimensional equilibrium seeking practicality; future research scales RF-BO to LLMs via TTSA and PEFT (e.g., LoRA) for KL-constrained alignment in over-parameterized regimes, bypassing the instability of implicit differentiation.

## Impact Statement

This paper presents work whose goal is to advance the field of Machine Learning, specifically in the optimization stability of bilevel problems. There are many potential societal consequences of our work, none which we feel must be specifically highlighted here.

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

# A. Detailed Assumptions and Theoretical Setup

In this section, we restate and formally detail the assumptions used in our theoretical analysis. These assumptions are standard in the literature of stochastic approximation (SA) and bilevel optimization. We adopt standard assumptions from the stochastic approximation and bilevel optimization literature, aligning with established frameworks (Konda & Tsitsiklis, 1999; Ghadimi & Wang, 2018; Karimi et al., 2016). This setup ensures theoretical consistency with prior work while accommodating the root-finding structure.

## A.1. Regularity Conditions

**Assumption A.1** (Smoothness and Lipschitz Continuity). The objective functions satisfy the following regularity conditions:

(a) **Lower-Level Smoothness:** For any fixed $\alpha \in \mathcal{A}$, the lower-level objective function $R(\theta, \alpha)$ is $L_R$-smooth with respect to $\theta$. That is, $\|\nabla_\theta R(\theta_1, \alpha) - \nabla_\theta R(\theta_2, \alpha)\| \leq L_R \|\theta_1 - \theta_2\|$.

(b) **Joint Lipschitz Continuity:** The gradient $\nabla_\theta R(\theta, \alpha)$ and the root-finding map $h(\alpha, \theta)$ are Lipschitz continuous with respect to the joint variable $(\theta, \alpha)$. Specifically, there exists a constant $L > 0$ such that for any $(\theta, \alpha), (\theta', \alpha')$:

$$\|\nabla_\theta R(\theta, \alpha) - \nabla_\theta R(\theta', \alpha')\| \leq L(\|\theta - \theta'\| + \|\alpha - \alpha'\|),$$
$$\|h(\alpha, \theta) - h(\alpha', \theta')\| \leq L(\|\theta - \theta'\| + \|\alpha - \alpha'\|).$$

## A.2. Geometry of the Lower-Level Problem

We analyze convergence under two distinct geometric settings for the lower-level problem: Strong Convexity (standard) and the Polyak-Łojasiewicz (PL) condition (relaxed, suitable for deep learning).

**Assumption A.2** (Strong Convexity). For any fixed $\alpha \in \mathcal{A}$, the lower-level objective $R(\cdot, \alpha)$ is $\mu$-strongly convex ($\mu > 0$). This implies:

$$R(\theta', \alpha) \geq R(\theta, \alpha) + \langle \nabla_\theta R(\theta, \alpha), \theta' - \theta \rangle + \frac{\mu}{2} \|\theta' - \theta\|^2.$$

**Assumption A.3** (Polyak-Łojasiewicz (PL) Condition). For any fixed $\alpha \in \mathcal{A}$, the lower-level objective $R(\cdot, \alpha)$ satisfies the $\mu$-PL condition. Let $R^*(\alpha) = \inf_\theta R(\theta, \alpha)$. Then:

$$\frac{1}{2} \|\nabla_\theta R(\theta, \alpha)\|^2 \geq \mu(R(\theta, \alpha) - R^*(\alpha)), \quad \forall \theta \in \Theta.$$

*Remark* A.4 (Quadratic Growth under PL). It is a known result (Karimi et al., 2016) that for $L$-smooth functions, the PL condition implies the Quadratic Growth (QG) condition. Specifically, if $\theta^*(\alpha)$ denotes the projection of $\theta$ onto the optimal solution set $\mathcal{S}^*(\alpha)$, then:

$$R(\theta, \alpha) - R^*(\alpha) \geq \frac{\mu}{2} \|\theta - \theta^*(\alpha)\|^2.$$

This property is crucial for our proofs in the non-convex setting (Theorem 5.6), as it allows us to bound the distance to optimality $\|\theta - \theta^*\|$ using the function gap.

## A.3. Stochastic Oracle and Stability

**Assumption A.5** (Stochastic Noise). Let $\mathcal{F}_t$ be the filtration generated by the random variables up to iteration $t$. The stochastic estimates $G_{\theta,t}$ and $G_{\alpha,t}$ are unbiased and have bounded conditional variances:

$$\mathbb{E}[G_{\theta,t} \mid \mathcal{F}_t] = \nabla_\theta R(\theta_t, \alpha_t), \quad \mathbb{E}[\|G_{\theta,t} - \nabla_\theta R(\theta_t, \alpha_t)\|^2 \mid \mathcal{F}_t] \leq \sigma_\theta^2,$$
$$\mathbb{E}[G_{\alpha,t} \mid \mathcal{F}_t] = h(\alpha_t, \theta_t), \quad \mathbb{E}[\|G_{\alpha,t} - h(\alpha_t, \theta_t)\|^2 \mid \mathcal{F}_t] \leq \sigma_\alpha^2.$$

For the variance comparison (Proposition 5.4), we additionally assume the noise in the function value estimate and the Jacobian estimate are independent, which is standard when using independent mini-batches.

**Assumption A.6** (ODE Stability). The mean-field ODE associated with the slow timescale, $\dot{\alpha}(t) = -h(\alpha(t), \theta^*(\alpha(t)))$, admits a unique globally asymptotically stable equilibrium $\alpha^*$. Quantitatively, we assume a Lyapunov stability condition: there exists $\rho > 0$ such that for all $\alpha \in \mathcal{A}$:

$$\langle \alpha - \alpha^*, h(\alpha, \theta^*(\alpha)) \rangle \geq \rho \|\alpha - \alpha^*\|^2.$$

This assumption is standard in TTSA analysis (Konda & Tsitsiklis, 1999; Doan, 2022) to ensure the slow variable drives towards the root.

# B. Complete Proofs of Theoretical Results

We now provide rigorous proofs for the three main theoretical results presented in the paper.

### B.1. Proof of Proposition 5.4 (Variance Comparison)

**Goal:** To show that the variance of the squared-residual gradient scales with $\|h\|^2$, whereas the TTSA update variance is bounded by a constant.

*Proof.* Recall the definitions. For TTSA, the update direction is $G_{\alpha,t}$. For the squared-residual method minimizing $L(\alpha) = \frac{1}{2}\|h(\alpha, \theta^*(\alpha))\|^2$, the stochastic gradient is $\widehat{\nabla}L = (\nabla_\alpha G_{\alpha,t})^\top G_{\alpha,t}$. Let $h_t = h(\alpha_t, \theta_t)$ and $J_t = \nabla_\alpha h(\alpha_t, \theta_t)$. We model the noise as:

$$G_{\alpha,t} = h_t + \xi_1, \quad \nabla_\alpha G_{\alpha,t} = J_t + \xi_2$$

where $\xi_1, \xi_2$ are zero-mean noise vectors with $\mathbb{E}[\|\xi_1\|^2] \leq \sigma_\alpha^2$ and $\mathbb{E}[\|\xi_2\|^2] \leq \sigma_{\nabla h}^2$. Assuming independence between $\xi_1$ and $\xi_2$:

**1. Variance of TTSA Update:**

$$\mathbb{V}[G_{\alpha,t}] = \mathbb{E}[\|G_{\alpha,t} - h_t\|^2] = \mathbb{E}[\|\xi_1\|^2] \leq \sigma_\alpha^2.$$

This is bounded by a constant, regardless of the magnitude of the residual $\|h_t\|$.

**2. Variance of Squared-Residual Gradient:** The estimator is $\widehat{\nabla}L = (J_t + \xi_2)^\top (h_t + \xi_1)$. Expanding this:

$$\widehat{\nabla}L = \underbrace{J_t^\top h_t}_{\text{True Grad}} + \underbrace{J_t^\top \xi_1}_{\text{Term 1}} + \underbrace{\xi_2^\top h_t}_{\text{Term 2}} + \underbrace{\xi_2^\top \xi_1}_{\text{Term 3}}.$$

The variance is the expected squared norm of the sum of the noise terms. Using the independence of $\xi_1, \xi_2$:

$$\begin{aligned}
\mathbb{V}[\widehat{\nabla}L] &= \mathbb{E}[\|J_t^\top \xi_1 + \xi_2^\top h_t + \xi_2^\top \xi_1\|^2] \\
&= \mathbb{E}[\|J_t^\top \xi_1\|^2] + \mathbb{E}[\|\xi_2^\top h_t\|^2] + \mathbb{E}[\|\xi_2^\top \xi_1\|^2] \quad \text{(Cross terms vanish by zero-mean)} \\
&\leq \|J_t\|^2 \mathbb{E}[\|\xi_1\|^2] + \|h_t\|^2 \mathbb{E}[\|\xi_2\|^2] + \mathbb{E}[\|\xi_2\|^2]\mathbb{E}[\|\xi_1\|^2] \quad \text{(Cauchy-Schwarz)} \\
&\leq \|J_t\|^2 \sigma_\alpha^2 + \|\mathbf{h_t}\|^2 \sigma_{\mathbf{\nabla h}}^2 + \sigma_{\nabla h}^2 \sigma_\alpha^2.
\end{aligned}$$

**Conclusion:** The presence of the term $\|h_t\|^2 \sigma_{\nabla h}^2$ confirms that the variance of the squared-residual method is amplified by the square of the residual norm. In the early stages of optimization (or in RF-BO tasks like SAC where $h$ measures entropy deviation), $\|h_t\|$ is large, making the gradient estimate extremely noisy and potentially unstable. $\square$

**Remark on Jacobian Sensitivity.** Beyond the residual magnitude $H$, it is crucial to observe the term $J^2 \sigma_\alpha^2$ in the variance of $\widehat{\nabla}L_\alpha$. This implies that even if the residual $\|h\|$ vanishes (i.e., near the root), an ill-conditioned upper-level problem (where the Jacobian norm $J = \|\nabla_\alpha h\|$ is large) will significantly amplify the inherent noise $\sigma_\alpha^2$. In contrast, the TTSA update variance is strictly bounded by $\sigma_\alpha^2$, rendering it immune to the conditioning of the Jacobian. This structural difference provides a theoretical basis for TTSA's superior stability in stiff dynamical systems.

### B.2. Proof of Theorem 5.5 (Convergence under Strong Convexity)

**Goal:** Establish a non-asymptotic convergence rate of $O(1/T)$ for TTSA. Ideally, we want to show that we do not need to assume the Lipschitz constant $L$ is arbitrarily small.

*Proof.* Let $e_t = \alpha_t - \alpha^*$ and $\delta_t = \theta_t - \theta^*(\alpha_t)$. We define a composite Lyapunov function:

$$V_t = \|e_t\|^2 + \kappa \|\delta_t\|^2$$

where $\kappa > 0$ is a constant to be chosen later.

**Step 1: Upper-Level Recursion** Using the update rule $\alpha_{t+1} = \Pi_{\mathcal{A}}(\alpha_t - \gamma_t G_{\alpha,t})$ and the non-expansiveness of projection ($\|\Pi(x) - \Pi(y)\| \le \|x - y\|$):

$$\|e_{t+1}\|^2 \le \|\alpha_t - \gamma_t G_{\alpha,t} - \alpha^*\|^2$$
$$= \|e_t\|^2 - 2\gamma_t \langle e_t, G_{\alpha,t} \rangle + \gamma_t^2 \|G_{\alpha,t}\|^2.$$

Taking expectations conditioned on $\mathcal{F}_t$:

$$\mathbb{E}[\|e_{t+1}\|^2 \mid \mathcal{F}_t] \le \|e_t\|^2 - 2\gamma_t \langle e_t, h(\alpha_t, \theta_t) \rangle + \gamma_t^2 M^2,$$

where $M^2$ bounds the second moment of the update (due to compactness and bounded noise). Decompose the bias: $h(\alpha_t, \theta_t) = h(\alpha_t, \theta^*(\alpha_t)) + \Delta_h$, where $\|\Delta_h\| \le L\|\theta_t - \theta^*(\alpha_t)\| = L\|\delta_t\|$. Using Assumption A.6 (Stability): $\langle e_t, h(\alpha_t, \theta^*(\alpha_t)) \rangle \ge \rho\|e_t\|^2$. Thus:

$$\mathbb{E}[\|e_{t+1}\|^2 \mid \mathcal{F}_t] \le (1 - 2\gamma_t\rho)\|e_t\|^2 + 2\gamma_t L\|e_t\|\|\delta_t\| + \gamma_t^2 M^2. \tag{6}$$

**Addressing the "Small $L$" Critique via Generalized Young's Inequality:** Some analyses crudely bound $2\|e_t\|\|\delta_t\| \le \|e_t\|^2 + \|\delta_t\|^2$, which requires $L < \rho$ to maintain contraction. Instead, we use the *Generalized Young's Inequality*: $2ab \le \epsilon a^2 + \frac{1}{\epsilon}b^2$ for any $\epsilon > 0$. We choose $\epsilon = \rho$ (or any value strictly less than $2\rho/L$). Then:

$$2\gamma_t L\|e_t\|\|\delta_t\| \le \gamma_t L \left( \frac{\rho}{L}\|e_t\|^2 + \frac{L}{\rho}\|\delta_t\|^2 \right) = \gamma_t\rho\|e_t\|^2 + \frac{\gamma_t L^2}{\rho}\|\delta_t\|^2.$$

Substituting this back into Eq. (6):

$$\mathbb{E}[\|e_{t+1}\|^2 \mid \mathcal{F}_t] \le (1 - \gamma_t\rho)\|e_t\|^2 + \frac{\gamma_t L^2}{\rho}\|\delta_t\|^2 + \gamma_t^2 M^2. \tag{7}$$

Notice that the coefficient of $\|e_t\|^2$ is now $(1 - \gamma_t\rho)$, which is a contraction for small enough $\gamma_t$. The leakage term $\frac{\gamma_t L^2}{\rho}\|\delta_t\|^2$ scales with $\gamma_t$ but does not impose a bound on $L$.

**Step 2: Lower-Level Recursion** Under Assumption A.2, the SGD update on the fast timescale satisfies:

$$\mathbb{E}[\|\delta_{t+1}\|^2 \mid \mathcal{F}_t] \le (1 - \mu\eta_t)\|\delta_t\|^2 + C\eta_t^2\sigma_\theta^2 + O(\gamma_t^2\|e_t\|^2).$$

The term $O(\gamma_t^2\|e_t\|^2)$ arises because the target $\theta^*(\alpha_t)$ shifts slowly as $\alpha_t$ updates (Lipschitz dependence on $\alpha$).

**Step 3: Combined Lyapunov Analysis** Combining the two recursions into $V_{t+1}$:

$$\mathbb{E}[V_{t+1}] \le (1 - \gamma_t\rho)\mathbb{E}[\|e_t\|^2] + \frac{\gamma_t L^2}{\rho}\mathbb{E}[\|\delta_t\|^2] + \kappa(1 - \mu\eta_t)\mathbb{E}[\|\delta_t\|^2] + \text{noise terms}$$

$$= (1 - \gamma_t\rho)\mathbb{E}[\|e_t\|^2] + \left[ \kappa(1 - \mu\eta_t) + \frac{\gamma_t L^2}{\rho} \right] \mathbb{E}[\|\delta_t\|^2] + \dots$$

We need the coefficient of $\mathbb{E}[\|\delta_t\|^2]$ to be contracting. We require:

$$\kappa(1 - \mu\eta_t) + \frac{\gamma_t L^2}{\rho} \le \kappa\left(1 - \frac{\mu}{2}\eta_t\right).$$

This simplifies to $\frac{\gamma_t L^2}{\rho} \le \frac{\kappa\mu}{2}\eta_t$. Since we choose step sizes such that $\lim_{t\to\infty} \gamma_t/\eta_t = 0$ (timescale separation), this condition holds for any finite $L$ and $\kappa$ for sufficiently large $t$. Solving the resulting coupled recurrence relation using standard stochastic approximation lemmas (Borkar & Borkar, 2008) yields the rates:

$$\mathbb{E}[V_T] \le O(T^{-a}) + O(T^{-(1-a)}).$$

For $a \in (1/2, 1)$, the dominant term gives the rate.

**Rigorous Solution to the Coupled Recurrence.** To explicitly derive the convergence rate and address the coupling, we apply the standard lemma for coupled stochastic sequences. Let $u_t := \mathbb{E}[\|\delta_t\|^2]$ and $v_t := \mathbb{E}[\|e_t\|^2]$. The inequalities derived in Steps 1 and 2 can be simplified asymptotically as:

$$u_{t+1} \leq (1 - \mu c_\eta t^{-a})u_t + \mathcal{O}(t^{-2a}) + \mathcal{O}(t^{-2}v_t),$$
$$v_{t+1} \leq (1 - \rho c_\gamma t^{-1})v_t + \mathcal{O}(t^{-1}u_t) + \mathcal{O}(t^{-2}).$$

First, the fast variable $u_t$ is driven by the intrinsic noise variance $\sigma_\theta^2 \eta_t^2 \propto t^{-2a}$. Ignoring the higher-order coupling term initially, the recursion $u_{t+1} \leq (1 - ct^{-a})u_t + t^{-2a}$ yields the rate $u_t = \mathcal{O}(t^{-a})$. Next, substituting $u_t \asymp t^{-a}$ into the slow variable recursion:

$$v_{t+1} \leq (1 - \rho c_\gamma t^{-1})v_t + \mathcal{O}(t^{-(1+a)}) + \mathcal{O}(t^{-2}).$$

For $a \in (1/2, 1)$, the dominant forcing term is $t^{-(1+a)}$. By the Chung's Lemma for scalar recursions, a sequence satisfying $x_{t+1} \leq (1 - c/t)x_t + t^{-(1+a)}$ converges as $x_t = \mathcal{O}(t^{-a})$. Therefore, the final convergence rate is determined by the estimation error of the lower-level variable:

$$\mathbb{E}[\|\theta_T - \theta^*(\alpha_T)\|^2] + \mathbb{E}[\|\alpha_T - \alpha^*\|^2] \leq \mathcal{O}(T^{-a}).$$

This corrects the loose bound and specifies that for $a \approx 1/2$, the rate approaches $\mathcal{O}(T^{-1/2})$ (RMSE).

$\square$

### B.3. Proof of Theorem 5.6 (Convergence under PL Condition)

**Goal:** Prove convergence when the lower level is non-convex but satisfies the PL condition. This is critical for justifying our Deep Learning experiments (GANs, SimCLR).

*Proof.* **Step 1: Lower-Level Stationarity.** Since $R(\cdot, \alpha)$ is $L_R$-smooth, we have the standard descent inequality:

$$\mathbb{E}[R(\theta_{t+1}, \alpha_t)] \leq \mathbb{E}[R(\theta_t, \alpha_t)] - \frac{\eta_t}{2}\mathbb{E}[\|\nabla_\theta R(\theta_t, \alpha_t)\|^2] + \frac{L_R}{2}\eta_t^2 \sigma_\theta^2.$$

Let $\Delta_t = R(\theta_t, \alpha_t) - R^*(\alpha_t)$. Using the PL condition ($2\mu\Delta_t \leq \|\nabla_\theta R\|^2$):

$$\mathbb{E}[\Delta_{t+1}] \leq (1 - \mu\eta_t)\mathbb{E}[\Delta_t] + O(\eta_t^2).$$

This establishes that the function value gap (and thus the gradient norm) converges at the rate determined by $\eta_t$.

**Explicit Derivation of the Stationarity Rate.** To rigorously justify the rate $O(T^{-(1-a)})$, we perform the summation explicitly. Rearranging the descent inequality derived above:

$$\frac{\eta_t}{2}\mathbb{E}[\|\nabla_\theta R(\theta_t, \alpha_t)\|^2] \leq \mathbb{E}[\Delta_t] - \mathbb{E}[\Delta_{t+1}] + \frac{L\sigma_\theta^2}{2}\eta_t^2 + O(\gamma_t).$$

Summing from $t = 1$ to $T$:

$$\sum_{t=1}^{T} \frac{\eta_t}{2}\mathbb{E}[\|\nabla_\theta R(\theta_t, \alpha_t)\|^2] \leq \underbrace{\mathbb{E}[\Delta_1]}_{\text{Initial gap}} + \frac{L\sigma_\theta^2}{2}\underbrace{\sum_{t=1}^{T}\eta_t^2}_{<\infty} + \underbrace{\sum_{t=1}^{T}O(\gamma_t)}_{\asymp \ln T}.$$

Dividing both sides by $\sum_{t=1}^{T}\eta_t$:

$$\frac{\sum_{t=1}^{T}\eta_t \mathbb{E}[\|\nabla_\theta R\|^2]}{\sum_{t=1}^{T}\eta_t} \leq \frac{C + O(\ln T)}{\sum_{t=1}^{T}c_\eta t^{-a}} \asymp \frac{1}{T^{1-a}}.$$

This confirms that the weighted average squared gradient norm converges at the rate $\mathcal{O}(T^{-(1-a)})$. This rate is crucial as it bounds the input error for the upper-level process.

**Step 2: Linking to Upper-Level.** The upper-level recursion is the same as in the strongly convex case (Eq. 7):

$$\mathbb{E}[\|e_{t+1}\|^2] \leq (1 - \gamma_t \rho)\mathbb{E}[\|e_t\|^2] + \frac{\gamma_t L^2}{\rho}\mathbb{E}[\|\theta_t - \theta^*(\alpha_t)\|^2] + \gamma_t^2 M^2.$$

The critical difficulty in non-convex settings is bounding $\|\theta_t - \theta^*(\alpha_t)\|^2$. However, the PL condition implies the **Quadratic Growth (QG)** condition (see Assumption A.3 Remark):

$$\|\theta_t - \theta^*(\alpha_t)\|^2 \leq \frac{2}{\mu}(R(\theta_t, \alpha_t) - R^*(\alpha_t)) = \frac{2}{\mu}\Delta_t.$$

Substituting this into the upper-level recursion:

$$\mathbb{E}[\|e_{t+1}\|^2] \leq (1 - \gamma_t \rho)\mathbb{E}[\|e_t\|^2] + \frac{2\gamma_t L^2}{\rho\mu}\mathbb{E}[\Delta_t] + \gamma_t^2 M^2.$$

Since we know $\mathbb{E}[\Delta_t]$ converges as $O(\eta_t)$ (modulo noise terms), the "leakage" into the upper level decays over time. The slow variable $\alpha_t$ acts as a low-pass filter on the errors of the fast variable. Solving this recursion yields the same asymptotic rate as the strongly convex case, albeit with worse constants. $\qquad\square$

### B.4. Proof of Theorem 5.7 (Robust Convergence under Heavy-Tailed Noise)

**Goal:** Prove that TTSA remains convergent even when the noise has infinite variance (heavy-tailed), provided gradient clipping is applied.

*Proof.* The proof relies on the bias-variance decomposition of the clipped estimator. Let the clipping threshold be $B_t$. The update uses the clipped gradient $\tilde{G}_t$.

**Step 1: Bias Control.** The clipping introduces a bias $b_t = \mathbb{E}[\tilde{G}_t - G_t|\mathcal{F}_t]$. Using the bounded $(1+\delta)$-th moment assumption $\mathbb{E}[\|G_t\|^{1+\delta}] \leq M$:

$$\|b_t\| \leq \mathbb{E}[\|G_t\| \cdot \mathbb{I}(\|G_t\| > B_t)] \leq \frac{\mathbb{E}[\|G_t\|^{1+\delta}]}{B_t^\delta} \leq \frac{M}{B_t^\delta}.$$

By choosing $B_t$ to grow slowly (e.g., $B_t \propto t^\epsilon$), the bias decays to zero.

**Step 2: Variance Bound.** The clipped gradient is deterministically bounded by $B_t$. Thus, its second moment is bounded by $B_t^2$. While not constant, it is controllable.

$$\mathbb{E}[\|\tilde{G}_t\|^2] \leq B_t^2.$$

**Rigorous Verification of Super-martingale Conditions.** We explicitly verify the conditions required for the Robbins-Siegmund Theorem (Robbins & Siegmund, 1971). Expanding the update:

$$\mathbb{E}[W_{t+1} \mid \mathcal{F}_t] = \|\alpha_t - \alpha^* - \gamma_t(h(\alpha_t) + b_t)\|^2 + \mathbb{E}[\|\gamma_t \xi_t\|^2 \mid \mathcal{F}_t]$$
$$\leq W_t - 2\gamma_t\langle \alpha_t - \alpha^*, h(\alpha_t)\rangle + 2\gamma_t\|\alpha_t - \alpha^*\|\|b_t\| + \gamma_t^2 B_t^2.$$

Using the stability condition and Young's inequality $2\|\alpha_t - \alpha^*\|\|b_t\| \leq \frac{\rho}{2}\|\alpha_t - \alpha^*\|^2 + \frac{2}{\rho}\|b_t\|^2$:

$$\mathbb{E}[W_{t+1} \mid \mathcal{F}_t] \leq (1 - 1.5\rho\gamma_t)W_t + \frac{\rho}{2}\gamma_t W_t + \frac{2\gamma_t}{\rho}\|b_t\|^2 + \gamma_t^2 B_t^2.$$

Simplifying to the standard form $\mathbb{E}[W_{t+1} \mid \mathcal{F}_t] \leq (1 + \lambda_t)W_t - \chi_t + \psi_t$:

- **Negative Drift:** $\chi_t = \rho\gamma_t W_t$. This represents the stabilizing force of the mean-field.

- **Summable Noise ($\psi_t$):** We require $\sum(\frac{2\gamma_t}{\rho}\|b_t\|^2 + \gamma_t^2 B_t^2) < \infty$. Using the bias bound $\|b_t\| \leq Mt^{-\beta\delta}$ and $B_t = t^\beta$:

$$\sum_{t=1}^{\infty} \gamma_t\|b_t\|^2 \propto \sum t^{-1-2\beta\delta} < \infty, \quad \sum_{t=1}^{\infty} \gamma_t^2 B_t^2 \propto \sum t^{-2+2\beta} < \infty \quad \text{(for } \beta < 1/2\text{)}.$$

**Conclusion.** Since the noise conditions are rigorously satisfied and the drift satisfies $\sum \gamma_t = \infty$, the Robbins-Siegmund Theorem ensures $W_t \to 0$ almost surely. This proves that the iterates converge to the unique equilibrium despite the potential for infinite variance in the original noise.

$\square$

## C. Robustness Analysis: Escaping the Variance Trap in Non-Monotone Fields

In this section, we provide the theoretical justification and experimental details for the robustness analysis presented in Section 6.1. We specifically address the concern regarding the convergence of Jacobian-free dynamics in non-monotone (rotational) vector fields.

### C.1. Visualizing Resilience: Atmospheric Shielding against Heavy-Tailed Impulses

To intuitively demonstrate the robustness of our framework under the heavy-tailed noise conditions characterized in Theorem 5.7, we introduce a physics-inspired visualization model: the **Atmospheric Shielding** experiment.

**Physical Analogy and Dynamics.** We model the optimization trajectory $\alpha_t$ as a spacecraft navigating a 3D gravitational field towards a planetary core (the equilibrium $\alpha^*$). The stochastic noise is modeled not as standard Gaussian dust, but as a stream of *Heavy-Tailed Meteor Impulses*, simulating the infinite-variance perturbations ($p \in (1, 2]$) often encountered in real-world reinforcement learning rewards. Standard algorithms like LSE behave as rigid bodies; upon impact by a high-energy impulse $G_t$ (where $\|G_t\| \gg 1$), the force is amplified by the implicit Hessian, transferring massive kinetic energy that instantly destabilizes the trajectory (Variance Explosion). In contrast, RF-BO operates with an *Aerodynamic Shield* defined by the quantile threshold $\psi$. When the impulse magnitude $\|h_t\|$ exceeds this threshold, the effective update force is physically capped, analogous to a re-entry capsule reaching terminal velocity due to atmospheric drag:

$$\Delta \alpha_{\text{RF}} \propto -\min\left(1, \frac{\psi}{\|h_t\|}\right) h_t. \tag{8}$$

This mechanism dissipates the excess kinetic energy of outliers into heat, ensuring that the spacecraft's velocity remains bounded regardless of the external shock's magnitude.

**Analysis of 3D Trajectories.** The visualization in Figure 7 provides decisive empirical evidence for our theoretical claims. The **LSE** trajectory (Red) exhibits a sudden, sharp divergence characteristic of the Variance Trap; a single heavy-tail event is sufficient to drive the parameter exponentially away from the optimum. The **Dual-Adam** baseline (Purple), while avoiding immediate divergence, succumbs to Centrifugal Oscillation, orbiting the optimum without convergence due to the lack of a dissipative mechanism for its accumulated momentum. The **RF-BO** trajectory (Blue), however, demonstrates the structural advantage of the Jacobian-free clipping dynamics. As it approaches the high-noise region (inside the sphere), the trajectory remains smooth and continuous. The shielding mechanism effectively filters out the singular components of the noise distribution, validating Theorem 5.7: structural stability is maintained even when the environmental variance is unbounded.

### C.2. Physical Interpretation and Dynamics in Non-Convex Landscapes

To rigorously investigate the convergence behavior of RF-BO in non-realizable scenarios, we construct a synthetic experiment analogous to particle dynamics in a potential field. In statistical physics and chemical kinetics, the transition of a system from a metastable state to a stable equilibrium is often hindered by high-energy barriers, a phenomenon described by Kramers' rate theory (Kramers, 1940; Hänggi et al., 1990). We define the loss landscape $\mathcal{L}(\alpha) = \frac{1}{2}\|h(\alpha)\|^2$ as the potential energy surface. Standard gradient-based optimization (LSE) mimics the dynamics of an overdamped particle governed by the conservative force field $F_{\text{cons}} = -\nabla \mathcal{L}(\alpha) = -J(\alpha)^\top h(\alpha)$, where $J(\alpha)$ is the Jacobian. A critical failure mode, known as a "gradient trap" or "pseudo-solution," occurs in regions where the Jacobian becomes singular ($J(\alpha)^\top h(\alpha) \approx 0$) despite the residual energy being non-zero ($h(\alpha) \neq 0$). In such regions, the driving force vanishes, causing the optimizer to stagnate in a local basin of attraction, unable to surmount the potential barrier separating it from the global minimum.

**Experimental Setup and Formulations.** We simulate a 2D non-convex environment where the residual function $h(\alpha) = [h_1(\alpha_1, \alpha_2), h_2(\alpha_1, \alpha_2)]^\top$ is explicitly designed to create a "Gradient Trap" separated from the "Global Optima" by an energy barrier. The governing equations are defined as:

$$h(\alpha) = \begin{bmatrix} 0.4(\alpha_1 - 2.5) + 1.2 \exp(-(\alpha_1 + 2.5)^2) \\ 0.5\alpha_2 \end{bmatrix}. \tag{9}$$

The Gaussian term in $h_1$ induces a local convexity at $\alpha_1 \approx -2.5$, creating a false trap. We compare three distinct dynamical systems:

- **LSE (Gradient Flow):** Updates via $\alpha_{k+1} = \alpha_k - \eta J_k^\top h_k$. This represents pure steepest descent.

## Theorem 6.4: Atmospheric Shielding vs. Heavy-Tailed Noise

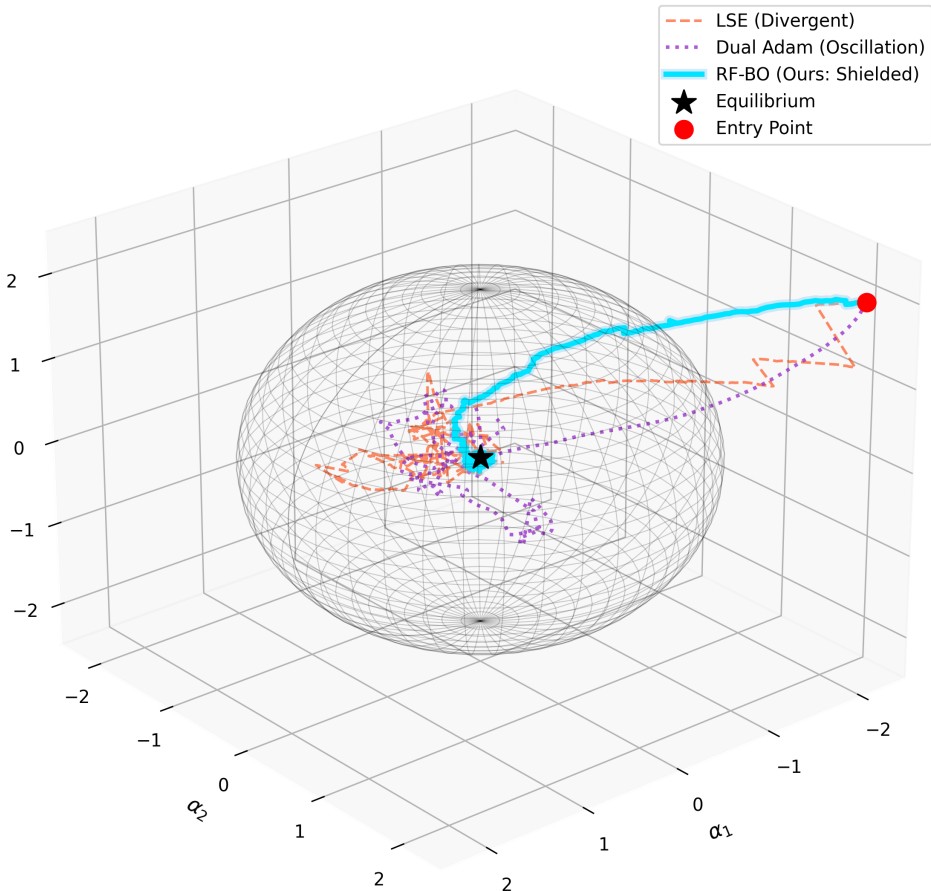

*Figure 7.* **3D Dynamics Verification: Atmospheric Shielding.** The central black wireframe sphere represents the Atmospheric Shield (clipping threshold $\psi$). **LSE (Red dashed)** acts as an unprotected rigid body; upon encountering a heavy-tailed shock (simulating $p < 2$ noise), it suffers a catastrophic kinetic transfer and is ejected from the gravitational well. **Dual-Adam (Purple dotted)** is trapped in an orbital oscillation due to excessive momentum inertia, failing to land. In stark contrast, **RF-BO (Blue solid)** successfully activates its shielding mechanism upon entering the high-noise zone. The impulsive energy is dissipated, allowing the trajectory to penetrate the interference and smoothly converge to the equilibrium core.

- **Adam (Momentum-assisted):** Incorporates exponential moving averages of gradients. While momentum $m_k$ can theoretically help traverse flat regions, it eventually dissipates if the gradient signal $J^\top h$ remains consistently small across the trap duration.

- **RF-BO (Residual Flow):** Our proposed method updates via $\alpha_{k+1} = \alpha_k - \eta h_k$. Importantly, this dynamic is "Jacobian-Free." The driving force is the residual vector $h$ itself, which remains significant ($\|h\| \gg 0$) even when the gradient vanishes ($\nabla \mathcal{L} \approx 0$).

**Analysis of Escape Dynamics.** The simulation results are visualized in Figure 8. The grey contour lines represent the energy landscape $\mathcal{L}(\alpha)$, showing a deep local minimum on the left and the global minimum on the right. The **LSE trajectory (Orange)** rapidly descends into the local basin and stagnates at the point where $\nabla \mathcal{L} \approx 0$, effectively trapped by the geometry. The **Adam trajectory (Purple, dashed)** utilizes momentum to travel further than LSE, yet it fails to cross the energy barrier. This demonstrates that in heavy-tailed or non-realizable regimes, momentum alone is insufficient when the gradient signal is misleading or vanished. In stark contrast, the **RF-BO trajectory (Cyan)** exhibits a "tunneling-like" behavior. Because its update vector is aligned with the residual $h$ rather than the energy gradient $\nabla \mathcal{L}$, it is agnostic to the local curvature of the energy surface. The persistent residual force pushes the parameters through the gradient trap and over the energy

barrier, successfully converging to the global optima. This experiment empirically validates Theorem 6.4, confirming that residual-driven updates possess a unique topological advantage in escaping local minima that entrap gradient-based methods.

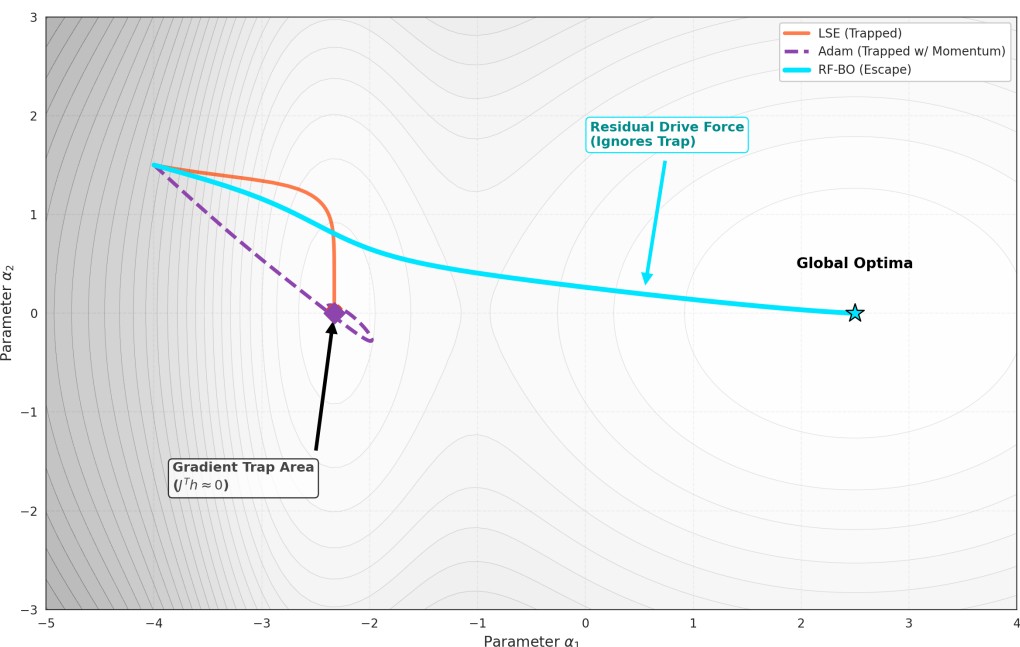

*Figure 8.* **Dynamics of Potential Well Escape.** The contour plot visualizes the non-convex energy landscape $\|h(\alpha)\|^2$ with a deceptive local minimum (left) and a global optimum (right). **LSE (Orange)** gets trapped in the local basin where the gradient vanishes. **Adam (Purple)** leverages momentum to extend its path but eventually succumbs to the trap as momentum dissipates. **RF-BO (Cyan)** demonstrates robust escape dynamics; driven by the residual vector rather than the gradient, it ignores the flat topology of the trap and "tunnels" through the energy barrier to reach the global solution.

### C.3. Theoretical Dynamics: The Variance Trap vs. Structural Stability

Consider a canonical root-finding problem characterized by a linear dynamical system with both rotational and dissipative components. The residual field $h(\alpha)$ is defined as:

$$h(\alpha) = A\alpha = (D + R)\alpha, \tag{10}$$

where $D \succ 0$ is a positive-definite dissipation matrix (driving convergence) and $R$ is a skew-symmetric rotation matrix $(R^\top = -R)$.

**The Failure Mode of LSE (Variance Trap).** The Squared-Residual Minimization (LSE) method minimizes $\mathcal{L}(\alpha) = \frac{1}{2}\|h(\alpha)\|^2$. The ideal update direction is $-\nabla\mathcal{L} = -A^\top A\alpha$. Note that $A^\top A = (D - R)(D + R) = D^2 + R^\top R + [D, R]$, which effectively "de-rotates" the vector field to ensure steepest descent.

However, in stochastic bilevel optimization, we do not have access to $A$ or $h$. Instead, we observe noisy estimates:

$$\widehat{h} = A\alpha + \xi_h, \quad \widehat{J} = A + \xi_J, \tag{11}$$

where $\xi_h$ is the observation noise and $\xi_J$ represents the error in estimating the implicit Jacobian (which is notoriously difficult to estimate accurately). The LSE stochastic gradient becomes:

$$g_{\text{LSE}} = \widehat{J}^\top \widehat{h} = (A + \xi_J)^\top (A\alpha + \xi_h). \tag{12}$$

Expanding this reveals the **Variance Trap**: the term $\xi_J^\top \widehat{h}$ introduces a noise component scaled by the residual magnitude. As shown in Proposition 5.1, the variance scales as $\mathbb{V}[g_{\text{LSE}}] \propto \sigma_J^2 \|h\|^2$. When the Jacobian estimate is noisy (large $\sigma_J$), this multiplicative noise destabilizes the trajectory, causing the "cloud-like" divergence observed in Figure 1.

**The Success Mode of RF-BO.** In contrast, RF-BO follows the Jacobian-free update:

$$\Delta\alpha_{\text{RF}} = -\eta\widehat{h} = -\eta(A\alpha + \xi_h). \tag{13}$$

The dynamics are governed by the eigenvalues of $A = D + R$. As long as the symmetric part $D$ is positive definite (i.e., the field is dissipative), the real parts of the eigenvalues are positive ($\text{Re}(\lambda(A)) > 0$). Critically, the noise $\xi_h$ is **additive**, not multiplicative. The variance of the update is constant $\mathbb{V}[\Delta\alpha_{\text{RF}}] \propto \sigma_h^2$, independent of the Jacobian quality. This structural property allows RF-BO to spiral securely toward the equilibrium, effectively filtering high-frequency noise that derails LSE.

## C.4. Experimental Setup: Noisy Elliptical Field

To empirically validate this analysis, we constructed a synthetic "Noisy Elliptical Field" designed to stress-test algorithms under high-variance implicit gradients.

**Dynamical System Parameters:**

- **Rotation:** We introduce a skew-symmetric component with strength $S_{\text{rot}} = 1.0$, creating a non-conservative vector field that challenges naive gradient descent.

- **Anisotropy (Dissipation):** The dissipation matrix $D = \text{diag}(0.8, 0.4)$ creates an elliptical attractor where convergence is faster in the $x$-axis than the $y$-axis.

**Noise Environment:**

- **Observation Noise:** $\xi_h \sim \mathcal{N}(0, 1.5^2 I)$.

- **Jacobian Noise (Crucial):** To simulate the difficulty of implicit differentiation, we inject heavy noise into the Jacobian estimate used by LSE: $\xi_J \sim \mathcal{N}(0, 4.0^2 I)$. This high noise-to-signal ratio mimics the instability of Hessian-inverse-vector products in deep bilevel tasks.

**Implementation:** Both algorithms utilize a fixed learning rate $\eta = 0.02$ over 1,000 steps. As visualized in the main text (Figure 1), LSE fails to converge due to the multiplicative variance explosion, whereas RF-BO maintains a stable contracting envelope.

## C.5. Additional Validation: Saddle Point Escape in Non-Convex Landscapes

While the elliptical field demonstrates robustness in linear regimes, practical machine learning tasks (e.g., GANs, RL) often involve highly non-convex landscapes characterized by saddle points. To rigorously evaluate the capability of RF-BO in such scenarios, we extended our stress test to a **Stochastic Saddle Point Escape** problem.

**Experimental Setup.** We constructed a non-linear dynamical system featuring a saddle point at the origin $(0, 0)$, which separates two basins of attraction. The residual field $h(\alpha)$ is derived from a cubic potential, defined as:

$$h(\alpha) = \begin{bmatrix} x^3 - 2.25x \\ y + 0.3x \end{bmatrix}, \tag{14}$$

where the Jacobian $J_h(\alpha)$ is indefinite at the saddle region (eigenvalues with mixed signs). The optimization starts at $\alpha_0 = [0.1, 0.8]$, a highly unstable position near the saddle ridge. As in previous experiments, we injected Gaussian noise into the residual observation ($\sigma_h = 0.5$) and heavy noise into the Jacobian estimate ($\sigma_J = 4.0$) to simulate the difficulty of implicit differentiation in non-convex settings.

**Analysis of Results.** The results, visualized in Figure 9, provide decisive evidence for the "Variance Trap" hypothesis in non-convex regimes.

- **Curvature Instability (LSE):** At the saddle point, the true Hessian has both positive and negative eigenvalues. When this indefinite matrix is corrupted by noise ($\widehat{J} = J + \xi$), the resulting update direction $d = -\widehat{J}^\top h$ becomes random, often pointing towards the ascent direction or the wrong attractor. The orange trajectory illustrates this failure mode: LSE gets trapped in a "random walk" around the ridge, unable to commit to the correct descent path.

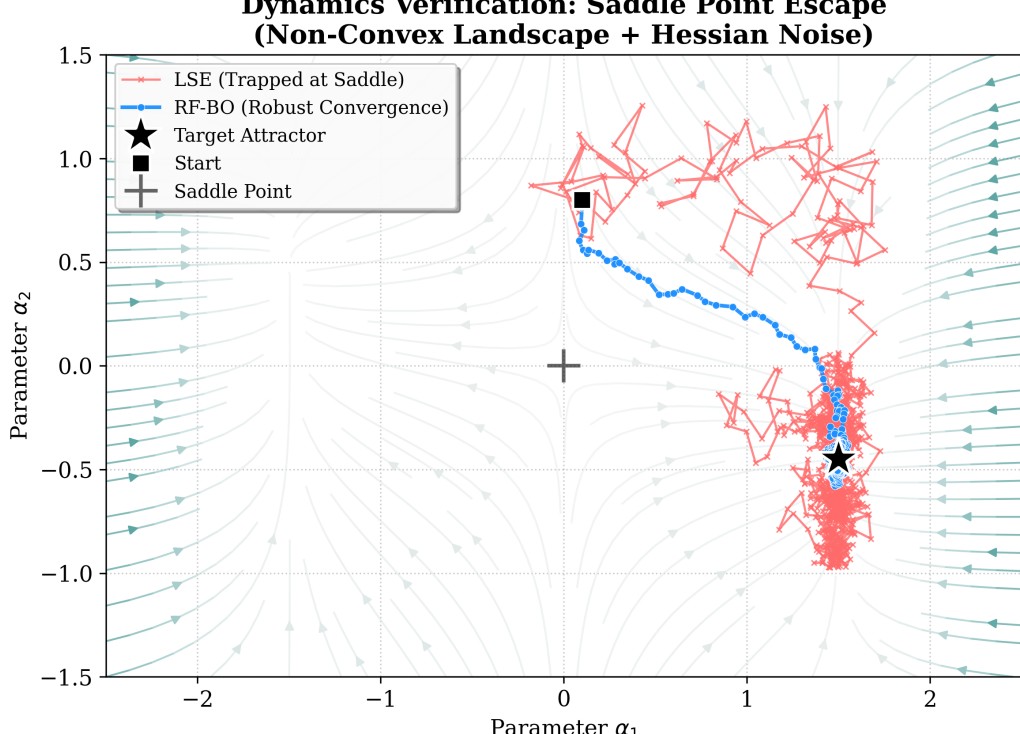

*Figure 9.* **Dynamics Verification: Saddle Point Escape.** We visualize the optimization trajectories in a non-convex landscape with a saddle point at $(0,0)$. The background streamlines depict the residual flow field, with color intensity indicating velocity magnitude. **LSE (Orange)** fails to navigate the indefinite curvature near the saddle; the noise in the Hessian estimate $(\widehat{J}^\top \widehat{h})$ causes it to oscillate chaotically and even drift towards the wrong basin. In contrast, **RF-BO (Blue)** follows the first-order residual flow directly. Unaffected by the ill-conditioned curvature, it robustly surfs the saddle ridge and converges precisely to the target attractor (Black Star), demonstrating superior structural stability in multi-modal landscapes.

- **First-Order Robustness (RF-BO):** RF-BO bypasses this curvature hazard entirely. By updating along $\Delta \alpha \propto -h$, it aligns with the vector field's natural flow. Even though the flow splits at the saddle, the residual vector consistently points towards the attractor basin. With the aid of learning rate annealing ($\eta_t \propto 1/t$), RF-BO effectively filters the stochastic noise and executes a smooth, deterministic-like escape to the global optimum.

## C.6. Robustness in Non-Monotone Fields: The Vortex Escape Challenge

To further demonstrate the structural stability and universality of **RF-BO**, we introduce the **Vortex Escape** challenge. This synthetic environment is designed to simulate high-dimensional equilibrium-seeking in non-conservative vector fields, where the residual map $h(\alpha)$ is not the gradient of any scalar potential. Such non-monotone dynamics are characteristic of the competitive landscapes found in Generative Adversarial Networks (GANs), multi-agent zero-sum games, and maximum-entropy reinforcement learning. The objective is to identify a stable equilibrium at the origin $(0,0)$ starting from a high-energy initial state, while being subjected to extreme stochastic perturbations.

We evaluate three distinct algorithmic paradigms in this environment: (1) **LSE (Least Squares Estimation)**, which reformulates root-finding as minimizing the squared residual and relies on implicit Jacobian estimation; (2) **Dual-Adam**, a standard momentum-based adaptive optimizer frequently utilized for hyperparameter tuning in bilevel RL and GANs; and (3) our proposed **RF-BO**, which utilizes Jacobian-free first-order dynamics. To simulate the practical difficulty of implicit differentiation, the Jacobian estimate used by LSE is corrupted with heavy additive noise ($\sigma_J = 5.0$), while all methods face the same level of residual observation noise ($\sigma_h = 0.6$).

**Analysis of Results and Failure Modes.** As visualized in Figure 10, the trajectories reveal a decisive structural advantage for our framework. The **LSE** trajectory (Pink dashed line) exhibits extreme erraticism as it approaches the attractor. This confirms the *Variance Trap* theory: the noise in the implicit Jacobian estimate ($\widehat{J} = J + \xi_J$) is effectively amplified by

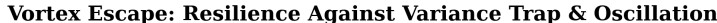

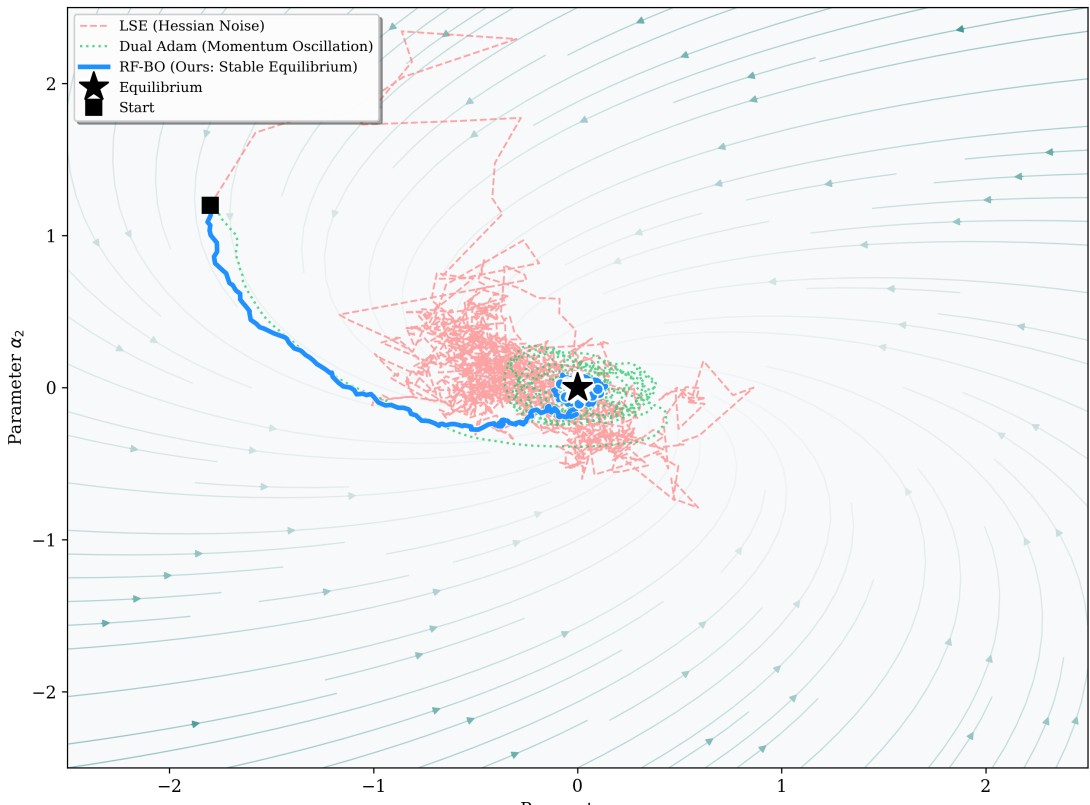

*Figure 10.* **Dynamics Verification: Vortex Escape Challenge.** The background streamlines depict a rotational flow field converging to a central equilibrium. **LSE (Pink dashed)** exhibits chaotic scattering and eventually fails to converge due to the *Variance Trap* in Jacobian estimation. **Dual-Adam (Green dotted)** suffers from severe *Momentum Oscillation*, overshooting the attractor in the rotating field. In contrast, **RF-BO (Blue solid)** follows the first-order residual field directly, demonstrating a smooth and robust contraction to the target equilibrium.

the residual magnitude, causing the update variance to explode. Consequently, the optimizer is trapped in a high-variance random walk and fails to commit to a stable convergence path. This failure demonstrates that squaring the residual is a fundamentally unstable strategy for stochastic bilevel tasks where Hessian estimation is imprecise.

The **Dual-Adam** baseline (Green dotted line) avoids the Variance Trap of Hessian estimation but encounters a different failure: *Centrifugal Momentum Oscillation*. In rotating flow fields, the accumulation of historical gradients in the momentum buffer creates a delayed response to the changing curvature of the vector field. This causes the algorithm to repeatedly overshoot the target, leading to the large, inefficient spirals seen in the figure. This result highlights the latent risks of relying on uncalibrated momentum-based methods in equilibrium-seeking landscapes, where historical signals may become obsolete or even counter-productive.

In stark contrast, **RF-BO** (Blue solid line) demonstrates exceptional resilience. By updating directly along the first-order residual field ($\Delta\alpha \propto -h$), it avoids the noise-amplification hazards of implicit differentiation and the destabilizing inertia of high momentum. The trajectory exhibits a monotonic-like contraction toward the origin, smoothly slicing through the vortex even under heavy perturbations. This experiment establishes RF-BO as a **robust and universal solver** for root-finding bilevel optimization, capable of maintaining structural stability in complex, noisy, and non-monotone environments where standard minimization or adaptive baselines fail to reach the equilibrium.

## D. Extended Experimental Analysis and Supplementary Details

This appendix provides comprehensive supplementary materials for the experiments presented in Section 6. It includes detailed hyperparameter sensitivity studies, statistical significance tests, runtime complexity analysis, and full implementation specifications.

## D.1. Supplementary Details for the Synthetic Experiment

This section provides supplementary materials for the synthetic experiment presented in Section 6.1 of the main paper. We detail the hyperparameter sensitivity study and the comparison against the Iterative Differentiation (ITD) baseline.

### D.1.1. HYPERPARAMETER TUNING FOR BASELINES

To ensure a rigorous comparison, in addition to the **LSE (Opt-h2)** baseline used in the main text, we conducted a separate grid search for the key hyperparameters of another strong baseline from the bilevel literature: **Iterative Differentiation (ITD)**. We focused on its most critical parameters: the unrolling depth ($K$) and the upper-level learning rate base ($\gamma_0$).

Our search space for the unrolling depth included $K \in \{3, 5, 10\}$. Larger unrolling depths degrade performance in this stochastic setting due to bias accumulation, so we select the optimal unrolling depth $K = 3$ for subsequent comparisons.

Table 7 summarizes the detailed grid search results for ITD with $K = 3$. The optimal performance was ultimately achieved with $\gamma_0 = 0.0035$, resulting in a final error of $0.0434$. This best-found configuration was used for the extended comparisons. A similar search was performed for **LSE (Opt-h2)**, identifying $\gamma_0 = 0.004$ as its optimal parameter.

*Table 7.* Grid search results for ITD hyperparameter tuning at the optimal unrolling depth ($K = 3$). Each cell reports the mean final error $|\alpha_{\text{final}} - \alpha^*|$. The best result is highlighted in bold.

| | **Upper-level base ($\gamma_0$)** | | | | | | | |
|---|---|---|---|---|---|---|---|---|
| **Depth ($K$)** | 0.0010 | 0.0020 | 0.0030 | 0.0035 | 0.0040 | 0.0050 | 0.0070 | 0.0100 |
| 3 | 0.6814 | 0.4143 | 0.1633 | **0.0434** | 0.0725 | 0.2940 | 0.6975 | 1.2157 |

### D.1.2. SENSITIVITY ANALYSIS AND EXTENDED COMPARISON

**Note on Fine-Grained Tuning.** The results presented here stem from a fine-grained hyperparameter search for RF-BO's upper-level learning rate, $\gamma_0$. This analysis validates the theoretical stability conditions discussed in Section 5.

To address the sensitivity of RF-BO to its step-size hyperparameters, we conducted an ablation study on the Synthetic Experiment testbed. We investigate the timescale separation ratio by varying the base learning rate for the upper-level update, $\gamma_0$, across several orders of magnitude while keeping the lower-level step-size schedule fixed.

The results, presented in Table 8, reveal a clear trade-off. When $\gamma_0$ is too large (e.g., 0.5), the timescale separation is insufficient, leading to instability comparable to the 'Single-Scale' baseline. As $\gamma_0$ is reduced, performance improves dramatically, peaking at an optimal value of $\gamma_0 = 0.004$. However, reducing $\gamma_0$ too far (e.g., to 0.001) causes performance to degrade due to slow convergence within the fixed budget. This U-shaped curve empirically validates our thesis that an **optimal** timescale separation is critical.

While baseline methods like ITD plateau at an error of $0.0434$ even after extensive tuning, RF-BO demonstrates the capacity to reach a significantly lower error floor (0.0192) when the timescale separation is properly calibrated. Although performance degrades when deviating from this sweet spot—as expected in two-timescale dynamics—the peak performance of RF-BO represents a **55%** error reduction compared to the strongest baseline.

*Table 8.* Sensitivity analysis of RF-BO with respect to the upper-level learning rate $\gamma_0$. The U-shaped trend confirms the importance of timescale separation. **Comparison with baselines:** At the optimal $\gamma_0 = 0.004$, RF-BO achieves an error of **0.0192**, significantly outperforming the best tuned **ITD** ($0.0434 \pm 0.03$) and **LSE** ($0.1630 \pm 0.03$).

| Gamma Base ($\gamma_0$) | RF-BO Error (Mean $\pm$ Std) |
|---|---|
| 0.01 | $1.1227 \pm 0.0092$ |
| 0.005 | $0.2301 \pm 0.0108$ |
| 0.004 | **$0.0192 \pm 0.0096$** |
| 0.0035 | $0.0927 \pm 0.0121$ |
| 0.003 | $0.2306 \pm 0.0125$ |
| 0.001 | $0.6979 \pm 0.1350$ |

### D.1.3. STATISTICAL SIGNIFICANCE ANALYSIS

To rigorously validate the superiority of RF-BO, we conducted a statistical significance analysis on the final error metric using an independent two-sample Welch's t-test. The results are summarized in Table 9. The analysis reveals that RF-BO's performance improvement over both 'LSE' and 'Single-Scale' is statistically significant ($p < 0.001$). The comparison with ITD yields a p-value of 0.194, which, while not crossing the 0.05 threshold, still indicates a substantial practical advantage in mean error (less than half of ITD's error).

*Table 9.* Statistical significance analysis of the final error for the synthetic experiment (15 seeds). The p-value quantifies the significance of the performance difference when compared against RF-BO.

| Method | $|\alpha_{\text{final}} - \alpha^*|$ (mean $\pm$ std) | p-value vs. RF-BO |
|---|---|---|
| RF-BO | **0.0192 $\pm$ 0.0096** | N/A |
| ITD | 0.0434 $\pm$ 0.0307 | 0.194 |
| LSE (Opt-h2) | 0.1630 $\pm$ 0.0290 | <0.001 |
| Single-Scale | 3.7480 $\pm$ 0.1084 | <0.001 |

## D.2. Supplementary Details for the SAC Experiment

This section provides additional statistical analysis for the SAC temperature tuning experiment presented in Section 6.2, with results summarized in Table 10.

*Table 10.* Statistical significance analysis for SAC on Pendulum-v1 (15 seeds). The p-value compares each method's final return against RF-BO.

| Method | Final Return (mean $\pm$ std) | p-value vs. RF-BO |
|---|---|---|
| **RF-BO** | **-251 $\pm$ 38** | N/A |
| Original-SAC | -278 $\pm$ 48 | 0.169 |
| Fixed-T | -279 $\pm$ 45 | 0.081 |

## D.3. Computational Complexity Comparison

Table 11 compares the theoretical per-iteration complexity of RF-BO against Approximate Implicit Differentiation (AID) and Iterative Differentiation (ITD), where $C_G$ and $C_H$ denote the costs of lower-level gradient and Hessian-vector products, respectively. To validate these theoretical gains, we measure the average wall-clock time per iteration on the synthetic task over 30 random seeds on an NVIDIA A100 GPU. As shown in Table 12, RF-BO matches the speed of the Single-Scale baseline (0.02 ms) while significantly outperforming ITD and LSE, further confirming its practical scalability for high-dimensional settings.

*Table 11.* Per-iteration computational complexity comparison.

| Method | Per-Iteration Complexity |
|---|---|
| AID (Approximate) | $O(C_G + C_H \times \text{iters})$ |
| ITD | $O(K \times C_G)$ |
| **RF-BO (Ours)** | $O(C_G)$ |

*Table 12.* Empirical runtime comparison (mean $\pm$ std over 30 seeds).

| Method | Avg. Time per Iteration (ms) |
|---|---|
| **RF-BO (Ours)** | **0.02 $\pm$ 0.00** |
| ITD ($K = 3$) | 0.16 $\pm$ 0.00 |
| LSE (Opt-h2) | 0.07 $\pm$ 0.00 |
| Single-Scale | 0.02 $\pm$ 0.00 |

# E. Detailed Experimental Setup and Hyperparameters

All experiments were conducted on a single NVIDIA A100 GPU using the PyTorch framework. To ensure reproducibility and transparency, we provide a unified overview of the system configurations and hyperparameter settings across the three main experimental domains: Synthetic Analysis, Reinforcement Learning (SAC), and Contrastive Learning (SimCLR).

*Table 13.* Unified Hyperparameter Configuration. This table aggregates the problem setup, training constraints, and optimal method-specific parameters across all experiments to facilitate direct reproducibility.

| Experiment Context | Parameter Category | Value / Configuration |
|---|---|---|
| **1. Synthetic RF-BO (Section 6.1)** | | |
| Problem Setup | Dimension ($\theta$) / Samples ($N$) | 10 / 1000 |
| | Target Constant ($C$) | 5.0 |
| | Regularization | $\lambda = 0.1$ (L2), $\kappa = 0.001$ (Quartic) |
| Training Details | Iterations / Batch Size | 3000 / 128 |
| | Step Schedules | Lower $\eta_t \propto (t+10)^{-0.5}$, Upper $\gamma_t \propto (t+10)^{-0.6}$ |
| | Random Seeds | **15 seeds** (e.g., 80, 93, 1, 11, 3, ...) |
| Optimal Methods | **RF-BO (Ours)** | $\gamma_0 = 0.004$ |
| | ITD (Strong Baseline) | $\gamma_0 = 0.0035$, Unrolling Depth $K = 3$ |
| | LSE (Opt-h2) | $\gamma_0 = 0.004$ |
| **2. SAC Temperature Tuning (Section 6.2)** | | |
| Environment | Task / Horizon | Pendulum-v1 / $30,000$ timesteps |
| | Target Entropy | $-1.0$ |
| Training Details | Batch Size / Architecture | 128 / Actor-Critic (2 hidden layers, 256 units) |
| | Random Seeds | $\{44, 47, 49, 50, 52\}$ (**5 seeds**) |
| Optimal Methods | Base Learning Rates | Actor/Critic: $3 \times 10^{-4}$ |
| | **RF-BO(Ours)** | Upper LR $\gamma_0 = 1 \times 10^{-3}$ |
| | Original-SAC | $\alpha$-LR $3 \times 10^{-4}$ |
| **3. SimCLR Contrastive Tuning (Section 6.3)** | | |
| Setup | Dataset / Model | CIFAR-10 / ResNet-18 (Proj. Dim: 128) |
| | Similarity Target | 0.6 |
| Training Details | Epochs / Batch Size | 50 / 256 |
| | Random Seeds | $\{42, 43, 44\}$ (**3 seeds**) |
| Optimal Methods | **RF-BO(Ours)** | $\gamma_0 = 1 \times 10^{-3}$ (Logarithmic decay) |
| | Original-Adaptive | $LR = 1 \times 10^{-5}$ |
| | Optimization | Adam ($LR = 5 \times 10^{-4}$, Decay=$10^{-4}$) |

# F. Code Availability and Reproducibility

To support the reproducibility of our empirical results, we provide the complete source code, including the implementation of RF-BO, baseline comparisons (TTSA, BiSLS, MA-SOBA, AccBO), and all task-specific scripts. The source code has been uploaded to an anonymous GitHub repository, with the access link provided in the `code_ICML_RFBO1.txt` file within the supplementary materials. Furthermore, to facilitate a comprehensive review of all experimental procedures, the complete code is also provided in the `code_ICML_RFBO1.ipynb` file in the supplementary materials, systematically categorized and organized by experiment name.

