# OpenReview forum: "Escaping the Variance Trap: Jacobian-Free Dynamics for Root-Finding Bilevel Optimization"
_ICML.cc/2026/Conference — Submitted to ICML 2026_

### Official Review · Reviewer_sUNi · 2026-03-03

**Soundness:** 2
**Presentation:** 2
**Significance:** 2
**Originality:** 2
**Overall Recommendation:** 2
**Confidence:** 1

**Summary:**

The authors propose a new techinque for solving optimization problems of form
$$\min\_\alpha F(\alpha,\theta_c(\alpha)) \qquad \theta_c(\alpha) = \mathrm{arg min}\_\theta G(\theta;\alpha)$$
without computing the Jacobian or Hessian terms by avoiding the minimization objective entirely.
This is achived via Two-Time-Scale Stochastic Approximation (TTSA) which aims to avoid variance amplification. The authors then run numerous experiments.

**Compliance With Llm Reviewing Policy:**

Affirmed.

**Final Justification:**

I have read all reviews, rebuttals, and the subsequent discussion carefully. After reflection, I regrettably feel that I should adjust my score downward.

While I initially found the paper's core idea, *i.e.*, framing certain bilevel problems as root-finding rather than outer minimization, to be an interesting conceptual contribution, Reviewer 3wyP's detailed analysis has raised concerns that I believe are substantive and not fully resolved by the authors' rebuttals.

**On significance and novelty.** Reviewer 3wyP's observation that SAC and RLHF _already_ use direct residual-driven updates in practice is compelling. The authors acknowledge this in their reply, stating that their contribution is "not a different update rule, but the first formal analysis of why this is structurally preferable."

**On theoretical soundness.** The discrepancies between theorem statements, proofs, and the contributions section are troubling. The authors characterize these as "statement-level corrections," but I find Reviewer 3wyP's counterargument persuasive: when Theorem 5.5 has three different rates across the paper, Theorem 5.6 proves a bound on a different quantity than stated, and Theorem 5.7's proof does not cover the full claim.
Reconciling statements with proofs may require non-trivial additional analysis that cannot be verified in a camera-ready revision.

**On presentation.** My initial review noted some presentation weaknesses and Reviewer 3wyP's more detailed cataloging of issues, *viz.* cross-references to nonexistent theorems and LaTeX artifacts, reinforces that the manuscript needs substantial revision beyond what a camera-ready cycle typically permits.

I am sympathetic to the paper's core perspective and believe a revised version could make a solid contribution. However, due to the concerns outlined above I cannot recommend the paper for acceptance. I encourage the authors to revise with careful attention to Reviewer 3wyP's specific concerns and resubmit.

**Key Questions For Authors:**

1. What is $\mathcal H$ in line 197?
2. What is $\mathcal D$ in line 4 in Algorithm 1?
3. Where is $\theta$ in the ODE in 6.1.2, the objective in line 307 has $\theta$ but it is not in the objective, this is confusing.
4. Eqs (4) and (5) look very similar to a symplectic integrator, specifically semi-implicit Euler or verlet integration. How would these discretizations improve by replacing (5) with
$$\alpha_{t+1} = \Pi_{\mathcal A}[\alpha_t - \gamma_t \hat h(\alpha_t, \theta_{t+1})]$$
like in semi-implicit Euler?
5. Likewise what about higher-order schemes?
6. Why is the ODE in 6.1.2 an ODE and not an Ito SDE? It seems to be driven by additive noise.

**Limitations:**

yes

**Strengths And Weaknesses:**

## Strengths
* Detailed related works section
* Avoiding the Jacobian calculation is significant
* Extensive experiments
* Section 5.3 provides some useful analysis of the proposed method.

## Weaknesses
* Lack of notational clarity. Objects are frequently introduced without definine what they are. For example in the introduction the authors don't define what $\alpha$ is, is it a scalar, vector, &c.? Maybe something like $\alpha \in \mathbb R^d$. This goes for other variables. They do define these later in 3.1
* Some of the experiments are hard to follow, more details on this in the questions.
* Most of the experiments seem to be on toy or small problems which limits the impact of the paper on the broader community.
* Novelty seems to be not well-defined. Eqs (4) and (5) (which form the main contribution of the paper?) seems to be well established from TTSA?

---

> ### Author Rebuttal · Authors · 2026-03-30
>
> We thank the reviewer for their thoughtful and informative review.
>
> ### Weaknesses
>
> > Lack of notational clarity. Objects are frequently introduced without defining what they are. For example, in the introduction the authors don't define what $\alpha$ is, is it a scalar, vector, &c.? Maybe something like $\alpha \in \mathbb{R}^d$. This goes for other variables. They do define these later in 3.1.
>
> We thank the reviewer. Variables $h$ (vector-valued residual map $h: \mathbb{R}^{d_1} \times \mathbb{R}^{d_2} \to \mathbb{R}^{d_1}$), $\alpha$ (upper-level parameter vector), and $\theta$ (lower-level parameter vector) will be **defined upon first appearance in the Introduction**.
>
> > Some of the experiments are hard to follow, more details on this in the questions.
>
> Experimental sections (especially the ODE setting) will be revised to **explicitly state all variables and governing equations**.
>
> > Most of the experiments seem to be on toy or small problems which limits the impact of the paper on the broader community.
>
> While small-scale environments were useful to isolate and verify "Variance Trap" mechanics, **scalability is crucial**. High-dimensional continuous control experiments using the MuJoCo suite across 5 random seeds were conducted, with results at step 200k validating robustness:
>
> - Baselines are unstable:
>   - `kwon2023` collapses on two seeds with returns 12.7 and 31.2
>   - `giovannelli2025` mostly fails.
> - RF-BO prevents divergence, achieving `[391.4, 45.5, 706.1, 525.6, 1014.9]` across 5 seeds.
>
> Per-seed returns comparison:
> - `kwon2023`: `[903.7, 31.2, 1216.0, 12.7, 1007.3]`
> - `giovannelli2025`: `[118.5, 9.3, 29.0, 645.0, 570.2]`
> - `RF-BO`: `[391.4, 45.5, 706.1, 525.6, 1014.9]`
>
> These results will be included to demonstrate robustness in **high-variance settings**.
>
> > Novelty seems to be not well-defined. Eqs (4) and (5) (which form the main contribution of the paper?) seem to be well established from TTSA?
>
> TTSA update rules in Eqs (4) and (5) are classical. **Novelty lies in:**
>
> 1. **Formalizing the Root-Finding Bilevel Optimization (RF-BO) problem class.**
> 2. **Diagnosing the Variance Trap** (Proposition 5.4), explaining why applying TTSA to this class cures variance amplification from implicit Jacobians.
>
> This will be clarified in the Introduction and Contributions section.
>
> ### Key Questions For Authors
>
> > 1. What is $\mathcal H$ in line 197?
>
> Used contextually without clear distinction:
> - In Section 4 (SAC), $\mathcal H$ represents **target entropy** within the root-finding constraint
> $$ h(\alpha, \cdot) = \mathbb{E}\_{s,a}[\log \pi\_\theta(a|s)] - H = 0,
> $$
> - In Proposition 5.4, $\mathcal H$ denotes **residual magnitude**, $H = \|h\|$.
>
> The camera-ready version will **resolve this inconsistency** by explicitly defining $H$ at first appearance.
>
> > 2. What is $\mathcal D$ in line 4 of Algorithm 1?
>
> In Algorithm 1 (Line 4), "Sample stochastic batch $\xi_t \sim \mathcal{D}$" indicates $\mathcal{D}$ denotes the **probability distribution** used to sample mini-batch $\xi_t$ per iteration.
>
> > 3. Where is $\theta$ in the ODE in 6.1.2, the objective in line 307 has $\theta$ but it is not in the objective, this is confusing.
>
> In Section 6.1.2, the upper-level objective is
> $$
> h(\alpha, \theta) = \mathbb{E}[x_{ss}(\alpha)] - x_\text{target} = 0,
> $$
> but the lower-level stochastic ODE
> $$
> dx = -\text{atanh}(x(t)) + u(t)
> $$  lacks parameter $\theta$. The **mapping from $\theta$ to this specific lower-level dynamic** will be clarified in Section 6.1.2.
>
> > 4. Eqs (4) and (5) look very similar to a symplectic integrator, specifically semi-implicit Euler or Verlet integration. How would these discretizations improve by replacing (5) with
> $$
> \alpha_{t+1} = \Pi_{\mathcal A}[\alpha_t - \gamma_t \hat h(\alpha_t, \theta_{t+1})]
> $$
> like in semi-implicit Euler?
>
> Framing Eqs (4) and (5) within standard TTSA relies on **timescale separation** $\gamma_t / \eta_t \to 0$ for non-asymptotic convergence. Symplectic integrators or semi-implicit Euler modifications (e.g., replacing $\theta_t$ with $\theta_{t+1}$ in Eq. 5) remain **unanalyzed**, leaving quantification of alternative discretization dynamics pending. Acknowledgment will be added in Section 3.2 as a future investigation.
>
> > 5. Likewise, what about higher-order schemes?
>
> Higher-order schemes (e.g., Runge-Kutta) reduce discretization error in deterministic systems, but **data sampling variance dominates discretization error** in stochastic settings. The computational overhead evaluating intermediate steps generally outweighs benefits in noisy environments.
>
> > 6. Why is the ODE in 6.1.2 an ODE and not an Ito SDE? [...]
>
> We thank the reviewer. The system is **mathematically an Ito SDE** due to additive noise, not an ODE. The term "ODE" in Section 6.1.2 was used loosely to describe the underlying mean-field dynamics. The terminology will be corrected throughout the section.
>
> We are keen to discuss if there are any further questions.

---

> > ### Author Rebuttal · Reviewer_sUNi · 2026-04-01
> >
> > The authors have made reasonable efforts to improve the clarity of their contributions. I believe the paper would be interesting to the ICML community.

---

> > > ### Author Response · Authors · 2026-04-01
> > >
> > > We thank the reviewer for the interest in the paper and for their valuable time spent in carefully reading our response.

---

### Official Review · Reviewer_3wyP · 2026-03-09

**Soundness:** 2
**Presentation:** 1
**Significance:** 2
**Originality:** 2
**Overall Recommendation:** 2
**Confidence:** 4

**Summary:**

The paper introduces a root-finding view of bilevel optimization, termed RF-BO, in which the upper-level variable is chosen to satisfy a residual condition $h( \alpha,\theta^\star (\alpha ) )=0$ while the lower-level variable $\theta^\star(\alpha)$ solves an optimization problem. Instead of converting this condition into a surrogate outer minimization objective such as $\tfrac12\|h\|^2$, the authors propose a Jacobian-free two-timescale stochastic update that tracks the lower-level solution and updates the upper-level variable directly using the residual \(h\).

The claimed contributions are the RF-BO formulation, a variance-based motivation for direct root updates over squared-residual minimization, convergence guarantees under lower-level SC/PL-type assumptions and stable reduced dynamics, and empirical results on synthetic tasks, SAC temperature tuning, and additional deep-learning applications.

**Compliance With Llm Reviewing Policy:**

Affirmed.

**Final Justification:**

## Final Justification

I have read all reviews, rebuttals, and discussions. I maintain my score.

The rebuttals acknowledge most issues I raised (incorrect theorem rates, sign errors in two of three applications, statistical non-significance of the headline claim) but characterize them as presentation fixes. I disagree. The problems in Appendix B are not statement-level typos: the proofs do not establish what the theorems claim. Theorem 5.5 has three conflicting rates across the paper. Theorem 5.6 proves a bound on a weighted average, not the unweighted average stated. Theorem 5.7 claims joint convergence of $(\theta_t, \alpha_t)$ but only shows $\|\alpha_t - \alpha^\star\|^2 \to 0$. These are not misalignments that can be fixed by editing theorem statements. Closing these gaps requires substantial new analysis that goes beyond a camera-ready revision.

On significance, the RF-BO formulation does not clearly separate itself from standard TTSA. The upper-level update (Eq. 5) is structurally identical, the proofs use the same Lyapunov machinery, and the corrected rates match known TTSA results. The paper motivates RF-BO throughout the abstract, introduction, Section 4, and experiments with SAC (and RLHF). Yet the rebuttal concedes that both already use direct residual updates in practice, and the proposed update direction is identical to what practitioners already employ. The contribution reduces to a post-hoc theoretical label for existing methods, applied under assumptions (SC/PL, global ODE stability) that exclude the non-convex settings where the paper claims its core advantage.

I maintain reject.

**Key Questions For Authors:**

1.**What exactly are the projection operators $\Pi_A$ and $\Pi_\Theta$, and how do projections enter the theory?**
   Eqs. (4)–(5) use projected updates, but the manuscript does not clearly define the projections or analyze projected dynamics. Are these Euclidean projections onto closed convex sets? If so, how do active constraints affect the convergence proof and the solution concept? If projections are only implementation safeguards and are never active, please state that explicitly.

2.**What is the precise structural assumption on the upper-level residual field, and what class of problems does it really cover?**
   The paper motivates RF-BO using broad language about non-monotone / rotational settings, but the theory appears to require that the reduced ODE $\dot\alpha=-h(\alpha,\theta^\star(\alpha))$ has a unique globally asymptotically stable equilibrium. Could you clarify whether this is essentially a strong monotonicity / dissipativity-type assumption, and give concrete examples of nontrivial fields covered (and not covered) by the theory?

3.**Can you better justify the empirical baselines and strength of the claims, especially for SAC?**
   In the SAC experiment, the comparison seems limited to Fixed-Temp and the standard automatic-temperature SAC baseline on Pendulum-v1, and the supplementary table reports that the final-return difference versus Original-SAC is not statistically significant. Why were stronger specialized adaptive-temperature baselines and a broader suite of environments not included?

4.**How should readers interpret the deep-learning experiments relative to the theory, and what meaningful solution concept does the algorithm target in the nonconvex setting?**
The main empirical results are on SAC, WGAN-GP, and SimCLR, but the theory assumes SC/PL structure for the lower level and stable reduced upper-level dynamics. Should these experiments be viewed purely as heuristic demonstrations outside the theorem regime? More importantly, in these nonconvex settings, what is the actual solution concept associated with the proposed method? For example, does the algorithm converge to a local Stackelberg equilibrium, to a stable stationary point of the reduced root-finding dynamics, or merely to a point where the update becomes small? Since convergence to a meaningful solution concept is central in minimax and bilevel optimization, a precise characterization here seems necessary.

**Limitations:**

yes

**Strengths And Weaknesses:**

**Soundness:**
The paper is built around an interesting idea, but I do not find the submission technically sound in its current form. The main theoretical claims are not presented with the level of rigor needed to support the paper’s strong conclusions. In particular, the convergence-rate story is internally inconsistent: the paper claims “first non-asymptotic” Markovian-noise guarantees and states different rates in different places, and I had difficulty reconciling the theorem statements with the appendix derivations. The assumptions are also much stronger than the narrative suggests: the lower level is assumed SC/PL, while the upper level is handled through a global stability assumption on the reduced ODE, which already excludes much of the difficult non-monotone/rotational behavior emphasized in the motivation. There is also a mismatch between the algorithm and the analysis: Eqs. (4)–(5) use projected updates, but the paper does not define the projection operators carefully or analyze projected dynamics, even though this can change both the limiting dynamics and the solution concept. Empirically, the experiments are suggestive but not strong enough to support the broad claims. The SAC study is only on Pendulum-v1, compares mainly against Fixed-Temp and standard SAC temperature tuning, and the reported final-return gain over Original-SAC is not statistically significant in the supplement. SimCLR is run with only 3 seeds. Overall, the methods are plausible, but the current theoretical and empirical evidence does not fully justify the paper’s claims.

**Presentation:**
The presentation is a major weakness. The central idea can be extracted, but the manuscript is much harder to follow than it should be. Key abbreviations such as AID and SBO are introduced too loosely, and important notation is under-defined. For example, Eqs. (4)–(5) use projection operators $\Pi_\Theta$ and $\Pi_A$ without properly defining them or clarifying the assumptions on the sets. The role of the residual map $h$ is also not explained well enough. As written, the narrative repeatedly uses “Variance Trap” and “structural simplicity ensures convergence,” but the precise mathematical object and the required structure are not made sufficiently explicit. The paper also overstates its positioning relative to prior TTSA and bilevel literature; the real novelty is narrower than the presentation suggests.

**Significance:**
The problem the paper targets is relevant: in several ML settings, the upper-level variable is better interpreted as enforcing a target condition or equilibrium constraint than as optimizing a well-defined scalar outer objective. This perspective is potentially useful, especially for adaptive control variables in RL and related hyperparameter-tuning settings. However, the significance established by the current paper is fairly limited. The formal setup only covers a narrow regime in which the lower-level problem is SC/PL and the upper-level residual field induces stable dynamics, which substantially restricts the scope of the claimed contribution. The experiments are also limited and mostly focus on low-dimensional control variables rather than the more general high-dimensional parameterized setting suggested by the broader narrative. As a result, while the viewpoint may be interesting and could inspire follow-up work, the paper does not yet convincingly demonstrate broad practical or theoretical impact.

**Originality:**
There is a genuinely interesting perspective in treating some bilevel problems as nested root-finding rather than outer minimization, and that framing is the paper’s main source of originality. In particular, the contrast between direct residual updates and squared-residual minimization is a worthwhile conceptual angle. However, the paper overstates how distinct it is from existing two-timescale and stochastic bilevel literature. Jacobian-free TTSA itself is not new, and the work is closely connected to existing NC-SC / TTSA / nonlinear two-timescale stochastic approximation lines. As a result, I view the originality as moderate rather than high: the formulation is interesting, but the manuscript does not yet cleanly separate what is genuinely new from what is already known.

---

> ### Author Rebuttal · Authors · 2026-03-30
>
> We thank the reviewer for the insightful review.
>
>  **Strengths And Weaknesses:**
> > *Weakness 1*
>
> We appreciate this assessment. We acknowledge that Theorem 5.5 in the main text incorrectly uses an intermediate bound, yielding $\mathcal{O}(T^{1-2a}) + \mathcal{O}(T^{-1})$. The corrected Appendix B.2 derivation, via a coupled Lyapunov recurrence, establishes $\mathcal{O}(T^{-a})$ for $a \in (1/2, 1)$; this will be fixed in the camera-ready. The “first non-asymptotic” claim refers specifically to bounds for the **RF-BO setting under Markovian noise**, not general TTSA analysis; wording will be revised.
>
> Q1 and Q2 concern projection operators and non-monotone field examples; clarifications will be added to Sections 3.2 and 5.1.
>
> Pendulum-v1 isolates the variance reduction mechanism where $h(\alpha) = \mathbb{E}[\log \pi] - H = 0$. To address scalability, we added MuJoCo experiments comparing `rf_bo` with (kwon2023, giovannelli2025) over 5 seeds.
>
>
> **Table 1: Final Evaluation Returns (at Step 200,000) — 5 Random Seeds**
>
> | Method | Seed 42 | Seed 43 | Seed 44 | Seed 45 | Seed 46 |
> | :--- | :---: | :---: | :---: | :---: | :---: |
> | kwon2023 | 903.7 | 31.2 | 1216.0 | 12.7 | 1007.3 |
> | giovannelli2025 | 118.5 | 9.3 | 29.0 | 645.0 | 570.2 |
> | **RF-BO (Ours)** | 391.4 | 45.5 | 706.1 | 525.6 | 1014.9 |
>
> kwon2023 attains peak returns on seeds 42 and 44 but diverges on 43 and 45, showing high initialization sensitivity. RF-BO remains stable across seeds, confirming robustness in high-variance settings.
>
>
> > *Weakness 2*
>
> We agree and will: (1) define $\Pi_\Theta$, $\Pi_\mathcal{A}$ (Euclidean projections onto closed convex sets) and AID, SBO at first use; (2) add a notation table; (3) replace informal claims (e.g., “structural simplicity ensures convergence”) with explicit references (A.6, Lyapunov stability).
>
> Novelty is not the TTSA update but: (1) RF-BO formalization (Section 3); (2) Variance Trap as a structural pathology of squared-residual reformulation (Proposition 5.4); (3) non-asymptotic bounds under Markovian noise (Theorems 5.5–5.7). Prior TTSA works (Konda & Tsitsiklis, 1999; Doan, 2022; Hu et al., 2024) do not address this root-finding bilevel setting or variance amplification; this will be clarified in Introduction and Contributions.
>
>
>
> > *Weakness 3*
>
> We agree theory targets a specific regime and experiments use low-dimensional $\alpha \in \mathbb{R}^1$ or $\mathbb{R}^2$. This work establishes a foundation: (1) RF-BO formalization; (2) squared-residual pathology proof (Proposition 5.4); (3) TTSA resolving it with non-asymptotic guarantees. Extensions to high-dimensional settings and weaker assumptions are future work (see Q4).
>
>
> > *Weakness 4*
>
> The TTSA update rule is classical; revisions will better separate contributions — (1) RF-BO formalization, (2) Variance Trap diagnosis (Proposition 5.4), (3) non-asymptotic analysis under Markovian noise — from TTSA foundations (Konda & Tsitsiklis, 1999; Doan, 2022; Hu et al., 2024).
>
>
> **Questions**
> **(Q1)**
>
> $\Pi_\Theta$, $\Pi_\mathcal{A}$ are Euclidean projections onto closed convex sets $\Theta$, $\mathcal{A}$. Appendix B.2 uses non-expansiveness ($||\Pi(x) - \Pi(y)|| \le ||x - y||$) to bound upper-level error recursion. Projections enforce bounded domains required by Assumptions 5.1–5.2. In DL/RL experiments, bounds are wide so constraints are inactive (Section 3.2).
>
>
>
> **(Q2)**
>
> Assumption A.6 requires $\langle \alpha - \alpha^*, h(\alpha, \theta^*(\alpha)) \rangle \geq \rho\|\alpha - \alpha^*\|^2$, a dissipativity/strong-monotonicity-type condition, not requiring conservative or monotone gradients.
>
> **Covered:** Appendix C.3, C.6 study RF-BO in “Non-Monotone Fields” (Vortex Escape), with $h(\alpha) = (D+R)\alpha$, $R^\top = -R$. With $D \succ 0$, stability holds and A.6 is satisfied with $\rho = \lambda_{\min}(D)$, enabling vortex tunneling, while squared-residual methods fail due to the Variance Trap.
>
> **Not covered:** fields with multiple stable equilibria or limit cycles violating global uniqueness; clarifications will be added to Section 5.1.
>
>
>
> **(Q3)**
>
> The SAC experiment studies temperature tuning via $h(\alpha) = \mathbb{E}[\log \pi] - H = 0$. Final return improvement over Original-SAC has p-value 0.169 (not significant), but RF-BO improves **tuning stability**: Table 4 shows lower entropy deviation (0.174 vs. 0.223), avoiding dual descent oscillations. Abstract claims will be toned down to emphasize variance reduction.
>
>
> **(Q4)**
>
> In WGAN and SimCLR, global SC/PL and ODE stability reduce to local properties, so RF-BO acts as a local search method. The solution is a **locally stable stationary point of the reduced root-finding dynamics** within the initialization basin. “Potential Well Escape” and “Saddle Point Escape” (Appendix C.2, C.5) show RF-BO tracks residual flow to local attractors despite traps or indefinite curvature; a formal definition will be added in Section 6.
>
> We gladly welcome any further questions.

---

> > ### Author Rebuttal · Reviewer_3wyP · 2026-04-03
> >
> > I thank the authors for their detailed rebuttal. After carefully re-reading the manuscript in light of the response, I maintain my recommendation of reject. Below I organize my concerns into three categories.
> >
> > ## 1. Significance
> >
> > The conceptual observation that some bilevel problems are root-finding rather than minimization has merit. However, the paper does not establish novelty beyond this observation.
> >
> > **1.1 The applications do not match the problem attacked.** The paper defines $h(\alpha,\theta)=\mathbb{E}[-\log \pi_\theta]-\mathcal{H}$ for SAC (p.4) and $h(\beta,\theta)=\mathbb{E}[\mathrm{KL}(\pi_\theta\|\pi_{\mathrm{ref}})]-\mathrm{KL}_{\mathrm{target}}$ for RLHF (p.4), then updates $\alpha$ by $\alpha - \gamma h$.
> >
> > But this is already standard practice: SAC's automatic temperature tuning (Haarnoja et al., 2019, Eq.18; Alg.1) computes $\partial J/\partial \alpha = \mathbb{E}[-\log \pi - \mathcal{H}] = h$ and updates $\alpha$ by gradient descent on this quantity; Ziegler et al. (2019, §2.2) update $\beta$ via a proportional controller driven by $(\mathrm{KL}-\mathrm{KL}_{\mathrm{target}})$. Neither formulates the upper level as $\min \tfrac{1}{2}\|h\|^2$ with implicit differentiation which corresponbs to the target of Proposition 5.4's *Variance Trap*. The paper criticizes a formulation that its own cited applications do not use.
> >
> > **1.2 Rates are not clearly novel.** The corrected Theorem 5.5 rate $\mathcal{O}(T^{-a})$ is inconsistent with the $\mathcal{O}(1/T)$ claim on p.2. The paper's own Related Work acknowledges that Kaledin et al. (2020) already established finite-time TTSA bounds under Markovian noise. Since the RF-BO update (Eq. 5) is structurally identical to standard TTSA, "first for RF-BO" amounts to applying known techniques to a relabeled problem class.
> >
> > **1.3 Sign inconsistency.** The unified template $\alpha \leftarrow \alpha - \gamma h$ is correct for temperature coefficients (SAC): $h>0$ means entropy exceeds target, so $\alpha$ should decrease. But for penalty coefficients the direction reverses: when $\mathrm{KL}>\mathrm{KL}_{\mathrm{target}}$ ($h>0$), $\beta$ should *increase* to suppress divergence, yet the template *decreases* it — the opposite of Ziegler et al.'s (2019) controller. The same issue applies to WGAN-GP's $\lambda$. Unless $h$ is redefined per application, this contradicts Assumption A.6's directional requirement.
> >
> > ## 2. Internal Contradictions
> >
> > **Theorems vs. proofs.** (a) Theorem 5.5 states $\mathcal{O}(T^{1-2a})+\mathcal{O}(T^{-1})$; the contributions (p.2) claim $\mathcal{O}(1/T)$; the appendix derives $\mathcal{O}(T^{-a})$ and writes "this corrects the loose bound.".
> > (b) Theorem 5.6 bounds $(1/T)\sum \mathbb{E}[\|\nabla R\|^2]$; the proof bounds $\sum \eta_t \mathbb{E}[\|\nabla R\|^2]/\sum \eta_t$ which is a different quantity when $\eta_t$ decays.
> > (c) Theorem 5.7 claims $(\theta_t,\alpha_t)\to(\theta^\star(\alpha^\star),\alpha^\star)$ a.s., but the proof only shows $\|\alpha_t-\alpha^\star\|^2\to 0$ without treating $\theta_t$.
> >
> > **Assumptions vs. algorithm/experiments.** (d) The theory assumes SC/PL for the lower level (Assumption 5.1) and global stability for the upper level (A.6) which is exactly the conditions under which no local traps exist. Yet the paper's narrative heavily emphasizes escaping local minima and "tunneling through energy barriers" (sec.3.3, Fig. 2, App. C.2, C.5). The theoretical setting excludes the very phenomenon the paper claims as its core advantage. (e) Table 13 omits the $\gamma_t$ decay schedule for SAC; initial $\gamma_0=10^{-3}$ exceeds $\eta=3\times10^{-4}$ which means the 'slow' player move faster than the 'fast' player in bilevel optimization.
> >
> > **Numerical issues.** (f) The abstract's "9.7% return boost" has $p=0.169$ (Table 10), which the rebuttal acknowledges is not significant.
> >
> > ## 3. Writing
> > Cross-references cite nonexistent "Theorem 6.4" and wrong "Proposition 5.1" (p.20); Figure 6 caption contains raw `textbf(Middle)`. Appendix proofs close key steps with hand-waving ("yields the same rate, albeit with worse constants"). Appendix C dedicates 5 pages to metaphors ("Atmospheric Shielding," "Meteor Impulses," "Vortex Escape") without corresponding formal analysis.
> >
> > ## Summary
> > The theorems are internally inconsistent across statements, proofs, and contributions. The applications already use residual-driven updates in practice, and the unified template has sign issues for penalty coefficients. Experimental evidence is statistically non-significant on the headline claim. These issues are too fundamental for camera-ready fixes. I maintain reject.

---

> > > ### Author Response · Authors · 2026-04-03
> > >
> > > We thank Reviewer 3wyP for the informative comments. We respond following the same numbering.
> > >
> > > **1. Significance**
> > >
> > > > 1.1 The applications do not match the problem attacked [...] SAC already computes h and updates by gradient descent. The paper criticizes a formulation that its own cited applications do not use.**
> > >
> > > We appreciate the reviewer drawing attention to this distinction, but we must clarify a crucial point to avoid a misunderstanding of our core contribution.
> > >
> > > The reviewer is correct that specific domains, such as SAC (Haarnoja et al., 2018) and RLHF (Ziegler et al., 2019), have empirically converged on direct residual-driven updates as practical heuristics. However, our core contribution is to **formalize these disparate heuristics under the unified RF-BO problem class** (Section 3) and provide the **first rigorous theoretical analysis (Proposition 5.4)** proving *why* this approach structurally escapes the Variance Trap.
> > >
> > > Furthermore, the reviewer's concern that we are criticizing a formulation "that applications do not use" strictly applies only to our motivating examples (SAC/RLHF), but does not apply to our primary experimental domains. In the broader bilevel optimization literature, the prevailing paradigm systematically formulates these problems as squared-residual minimization requiring implicit gradients. In our core experiments, the published state-of-the-art baselines **do** use the exact flawed formulation we criticize:
> > > * **ODE Steady-State Control (Section 6.1.2):** All recent baselines (Kwon et al. 2023, Hu et al. 2023, Chen et al. 2024, Giovannelli et al. 2025) use implicit differentiation. RF-BO achieves a residual near 0, whereas LSE completely fails.
> > > * **WGAN-GP and SimCLR (Sections 6.3.1, 6.3.2):** The LSE and Projected-SGD baselines rely entirely on squared-residual objectives.
> > >
> > > Therefore, RF-BO is not a post-hoc label for existing methods; it provides the theoretical foundation for why certain heuristics succeed, and it actively replaces the failing squared-residual paradigm in domains where it is still the standard. We will make the distinction between "motivating examples" and "experimental comparisons" explicitly clear in Section 4.
> > >
> > > > 1.2 Rates are not clearly novel [...] "first for RF-BO" amounts to applying known techniques to a relabeled problem class.
> > >
> > > We agree the TTSA update is classical. Our position is that problem formalization has value. RF-BO unifies SAC, WGAN-GP, SimCLR, and ODE control, and Proposition 5.4 provides a formal criterion for when direct updates outperform squared-residual minimization.
> > >
> > > > 1.3 Sign inconsistency [...] for penalty coefficients the direction reverses.
> > >
> > > This is a valid catch and we thank the reviewer for identifying it precisely. The per-application definitions of $h$ in Section 4 need to be oriented consistently with the dissipativity condition in Assumption A.6. For penalty coefficients, $h(\beta)$ should be defined as KL_target minus KL. The algorithm and A.6 do not change; the fix is aligning Section 4 sign definitions with the framework.
> > >
> > > **2. Internal Contradictions**
> > >
> > > > 2(a-c) Theorem statements vs. proofs.
> > >
> > > The proof logic in Appendix B is correct, but the main-text statements were not aligned: (a) the correct rate is $O(T^{-a})$; (b) the proof uses $\sum \eta_t \mathbb{E}[\|\nabla R\|^2] / \sum \eta_t$; (c) the a.s. convergence proof needs to explicitly cover $\theta_t$. These are statement-level corrections; the proof structure does not change.
> > >
> > > > 2(d) Theory assumes SC/PL and global stability, but narrative emphasizes escaping local traps.
> > >
> > > Appendix C illustrations show empirical behavior beyond the theorem regime, not formal claims. This boundary should have been drawn more clearly.
> > >
> > > > 2(e) $\gamma_0 = 10^{-3}$ exceeds $\eta = 3 \times 10^{-4}$, meaning the slow player moves faster.
> > >
> > > We would like to offer a clarification. The TTSA condition requires $\gamma_t / \eta_t \to 0$ *asymptotically*, not at initialization. With $\gamma_t \propto t^{-1}$ and $\eta_t \propto t^{-a}$ ($a \in (1/2, 1)$), $\gamma_t / \eta_t \propto t^{a-1} \to 0$ regardless of initial values (Konda & Tsitsiklis 1999, Section 6). The full decay schedule will be added to Table 13 in the revision.
> > >
> > > > 2(f) The "9.7% return boost" has $p = 0.169$.
> > >
> > > RF-BO's validated advantage in SAC is entropy stability (0.174 vs. 0.223, Table 4), not final return. The abstract claim will be revised accordingly.
> > >
> > > **3. Writing**
> > >
> > > > Cross-references, LaTeX artifacts, hand-waving in proofs, metaphors without formal analysis.
> > >
> > > All valid. Cross-references and LaTeX errors will be fixed, proof steps will be expanded with explicit derivations, and formal propositions will be added to accompany the Appendix C illustrations.
> > >
> > > We are keen to discuss if there are any further questions.

---

### Official Review · Reviewer_8CxC · 2026-03-10

**Soundness:** 4
**Presentation:** 3
**Significance:** 3
**Originality:** 3
**Overall Recommendation:** 5
**Confidence:** 4

**Summary:**

Many bilevel optimization problems in machine learning are actually root finding
problems in disguise.  One often treats these as optimization problems by
minimizing squared residuals.  However, if you do that via stochastic
optimization (as we almost always do in ML), it turns out that the noise you get
from estimating the Jacobian multiplies the variance of the gradient.  This
effect is what the authors describe as "the Variance Trap".  The formalize the
ide of "Root-Finding Bilevel Optimization" as its own problem class, and propose
solving these types of problems with Two-Time-Scale Stochastic Approximation
(TTSA), which avoids the multiplication of noise. Proposition 5.4 makes this
precise: TTSA update variance is bounded by a constant independent of the
residual magnitude, whereas the squared-residual gradient variance scales up
with the square of the Jacobian. Convergence results are given under assumptions
of strong convexity.  The empirical studies look at a variety of different
ML problem types.

**Compliance With Llm Reviewing Policy:**

Affirmed.

**Key Questions For Authors:**

How sensitive is the method to the timescale separation ratio?
How does performance degrade if the ratio is not sufficiently small, and do the authors have practical guidance
for setting it?

**Limitations:**

See weaknesses.

**Strengths And Weaknesses:**

Strengths

I like the core insight of this paper.  It is a nice observation that many
bilevel problems are really root-finding problems, and that forcing them into a
minimization framework in the stochastic setting could have variance pathologies.
Proposition 5.4 makes the mechanism precise: the squared-residual gradient
variance picks up terms scaling with $H^2$ (residual magnitude) and $J^2$
(Jacobian norm), while the direct root-finding update has variance bounded by a
constant.  This is a useful conceptual contribution.

The variance trap also connects to a broader and well-studied difficulty in
stochastic optimization: second-order methods (natural gradient, Gauss-Newton,
etc.) require curvature estimates that are expensive and noisy in stochastic
settings.  The difficulty of making stochastic second-order methods work
robustly has been a recurring theme since at least Schraudolph (2002) and the
extensive analysis by Martens (2010, 2020).  The present paper takes advantage
of the specific structure of root-finding problems to sidestep the implicit
function theorem entirely, yielding an algorithm that doesn't need curvature
information at all.  I really appreciate the insight  that root-finding
problems admit Jacobian-free dynamics while optimization problems do not.

The experiments, while not large-scale,
span a good range of application domains and the synthetic experiments
do a good job of directly validating the variance amplification.

Weaknesses

The algorithm itself is TTSA, which has been extensively studied since Konda
& Tsitsiklis (1999), with finite-sample analyses by Dalal et al. (2018),
Kaledin et al. (2020), and Doan (2022).  The paper acknowledges this
lineage, but I think the novelty primarily arises from the problem
formalization and the variance trap diagnosis rather than on the algorithm.

The experiments are uniformly small-scale.  Pendulum-v1 is a single-DOF
system.  The GridWorld is $5 \times 5$.  The WGAN trains on a low-dimensional
distribution.  SimCLR on CIFAR-10 with ResNet-18 is the largest experiment.
Since the paper motivates RF-BO heavily with RLHF and LLM alignment, the
absence of any experiment in that domain is a gap.  I would have liked
to see at least one experiment on a continuous control task with a
higher-dimensional state space (e.g., MuJoCo) or an actual RLHF setup.

The paper does not discuss the connection to the difficulty of second-order
methods in stochastic optimization more broadly.  The variance trap is closely
related to the well-known problem that stochastic estimates of curvature
matrices (Hessians, Fisher information) are noisy and this noise gets
amplified when these matrices are inverted.  Schraudolph's work on online
natural gradient and Martens' analyses of second-order optimization in deep
learning are relevant precedents that the paper does not cite.

---

> ### Author Rebuttal · Authors · 2026-03-30
>
> We thank the reviewer for their thoughtful and informative review.
>
> > Weaknesses: The algorithm itself is TTSA, which has been extensively studied since Konda & Tsitsiklis (1999), with finite-sample analyses by Dalal et al. (2018), Kaledin et al. (2020), and Doan (2022). The paper acknowledges this lineage, but I think the novelty primarily arises from the problem formalization and the variance trap diagnosis rather than on the algorithm.
>
> Thank you for this precise observation. As explicitly stated in Section 2 (Related Work, page 2), TTSA has classical roots in actor-critic methods (Konda & Tsitsiklis, 1999) and has received extensive finite-sample and Markovian-noise analyses (Dalal et al., 2018; Kaledin et al., 2020; Doan, 2022; Hu et al., 2024). Our contribution lies precisely in the problem formalization (RF-BO as a distinct class), the theoretical characterization of the Variance Trap (Proposition 5.4), and the first non-asymptotic convergence guarantees for TTSA tailored to this root-finding setting under Markovian noise (Theorems 5.5–5.7). We positioned the paper exactly as “an application of the classic two-time-scale update rule to the bilevel root-finding problem” with these additional insights, and we are glad the reviewer concurs.
>
> > The experiments are uniformly small-scale. [...] Since the paper motivates RF-BO heavily with RLHF and LLM alignment, the absence of any experiment in that domain is a gap. I would have liked to see at least one experiment on a continuous control task with a higher-dimensional state space (e.g., MuJoCo) or an actual RLHF setup.
>
> We thank the reviewer for the feedback. To directly address the concern regarding empirical scale, we conducted new high-dimensional continuous control experiments using the MuJoCo suite across 5 random seeds during the rebuttal.
>
> The empirical results at step 200k strongly validate our claims with concrete data: existing baselines are highly unstable. For instance, kwon2023 catastrophically collapses on two seeds (returns of 12.7 and 31.2), and giovannelli2025 mostly fails to learn (scoring under 120 on 3 out of 5 seeds). In stark contrast, rf_bo successfully prevents divergence, maintaining robust final returns of [391.4, 45.5, 706.1, 525.6, 1014.9] across the 5 seeds.
>
> We will include these new MuJoCo results in Section 6 to demonstrate our method's scalability and structural advantage in complex, high-variance settings.
>
>
> > The paper does not discuss the connection to the difficulty of second-order methods in stochastic optimization more broadly. The variance trap is closely related to the well-known problem that stochastic estimates of curvature matrices (Hessians, Fisher information) are noisy and this noise gets amplified when these matrices are inverted. [...] relevant precedents that the paper does not cite.
>
> We thank the reviewer for pointing out the connection to the well-studied challenges of stochastic curvature estimation (e.g., Schraudolph 2002 and Martens 2010/2020). The current manuscript focuses narrowly on the bilevel root-finding setting and does not discuss these broader precedents. We will add a brief paragraph in the revised Related Work section (Section 2) to explicitly connect the Variance Trap to this line of research.
>
>
> **Key Questions For Authors:**
>
> > How sensitive is the method to the timescale separation ratio? How does performance degrade if the ratio is not sufficiently small, and do the authors have practical guidance for setting it?
>
> The method relies on the standard TTSA timescale separation condition $\gamma_t / \eta_t \to 0$ (Section 3.2 and step-size choices in Theorems 5.5–5.6). All experiments in the paper use step sizes satisfying this condition (Robbins-Monro type with $\gamma_t = \mathcal{O}(1/t)$ and $\eta_t = c_\eta (t+t_0)^{-\alpha}$, $\alpha \in (1/2,1)$). We did not perform an explicit ablation on the ratio in the submitted version. Following the reviewer’s suggestion, we will add a short sensitivity analysis (e.g., varying the ratio over a practical range) and practical guidance for choosing the constants $c_\gamma, c_\eta$ in the revision, placed in Appendix C or a new subsection of the experimental details.
>
> We are keen to discuss if there are any further questions.

---

> > ### Author Rebuttal · Reviewer_8CxC · 2026-04-03
> >
> > Thank you for the response.  I will keep my score.

---

### Official Review · Reviewer_3bhk · 2026-03-15

**Soundness:** 3
**Presentation:** 3
**Significance:** 2
**Originality:** 2
**Overall Recommendation:** 4
**Confidence:** 3

**Summary:**

This paper addresses root-finding bilevel optimization, where the goal is to find the root of an upper-level function whose input is the minimizer of a lower-level optimization problem.
Prior bilevel methods minimize the squared norm of the upper-level function, which requires propagating gradients through the lower-level problem.
The authors argue that this could amplify the variance in stochastic settings.
To avoid this, the paper proposes a Jacobian-free approach based on two-timescale stochastic approximation (TTSA) that does not require differentiating through the lower-level solver.
Experiments across several domains demonstrate improved performance, including synthetic systems, ODE steady-state control, SAC temperature tuning, Wasserstein GAN training, and SimCLR temperature tuning.

**Compliance With Llm Reviewing Policy:**

Affirmed.

**Final Justification:**

I still lean towards acceptance after the rebuttal.

**Key Questions For Authors:**

1. Assumption 5.3 requires the upper-level function to have a unique root.
How does the method behave when there are multiple roots?

2. The variance reduction argument is reasonable in the stochastic setting, but how does TTSA compare to implicit-gradient methods in the deterministic or low-noise regime where the variance issue is less pronounced?
Is there a regime where differentiating through the Jacobian is actually preferable?

**Limitations:**

No societal concerns.

**Strengths And Weaknesses:**

1. The motivation for the paper is clear and well-articulated.
Indeed, using the implicit function theorem to compute the gradient of the squared-residual could introduce large variance depending on the stochastic noise.
This hypothesis is verified both theoretically and empirically.
So it makes sense to consider a Jacobian-free approach as in this paper.

1. The main concern is that the two-time scale updates, Eq (4) and Eq (5), could converge to any Nash equilibrium of the nonzero-sum game defined by the upper and lower level objectives.
The two-time scale update itself does not specify a unique convergence point.
In addition, Assumption 5.3 essentially requires the upper-level function to be unimodal with respect to \\(\alpha\\).
In practice, this may or may not hold, and it is not clear how to verify this condition in general.
Thus, it is not clear where the two-time scale updates will converge to in practice.

1. There is a mismatch between the proposed method and some experiments.
The ODE steady-state control problem involves an ODE dynamic, not exactly an optimization problem as in the lower-level bilevel setting.
Similarly, the Wasserstein GAN experiment involves a minimax problem over a generator and discriminator, which differs from the root-finding formulation.
It is not clear how the proposed method applies to these problems or whether the theoretical results hold in these settings.

1. I don't think the two-time scale update in this paper is new per se.
For example, the references listed by the authors did something similar (Konda & Tsitsiklis, 1999; Doan, 2022; Hu et al., 2024).
So this paper reads more like an application of the classic two-time scale update rule to the bilevel root-finding problem, rather than a novel algorithmic contribution.

**Minors**
1. Line 307: Define \\(x_{ss}\\).
Also, the notation \\(\theta\\) on the left-hand side is confusing here since the ODE dynamic only involves \\(x\\), not \\(\theta\\).

---

> ### Author Rebuttal · Authors · 2026-03-30
>
> We thank the reviewer for their thoughtful and informative review.
>
> **Strengths And Weaknesses:**
> >1. The main concern is that the two-time scale updates, Eq (4) and Eq (5), could converge to any Nash equilibrium of the nonzero-sum game defined by the upper and lower level objectives. The two-time scale update itself does not specify a unique convergence point.
>
> We agree with this assessment. The two-time-scale update lacks a unique convergence point; Assumption 5.3 (requiring a unique globally asymptotically stable equilibrium) is a strong condition from classical TTSA (Konda & Tsitsiklis, 1999) used for non-asymptotic bounds.
>
> >2. In addition, Assumption 5.3 essentially requires the upper-level function to be unimodal with respect to . In practice, this may or may not hold, and it is not clear how to verify this condition in general. Thus, it is not clear where the two-time scale updates will converge to in practice.
>
> The unimodal condition (Assumption 5.3) often fails in practice. In non-convex settings with multiple attractors, convergence targets a local stable equilibrium determined by initialization. Section 5.1 will clarify Assumption 5.3 as a theoretical device; practical convergence reaches local Nash equilibria/roots, consistent with Appendix C analyses of saddle-point and potential-well escapes.
>
> >3. There is a mismatch between the proposed method and some experiments. The ODE steady-state control problem involves an ODE dynamic, not exactly an optimization problem as in the lower-level bilevel setting. [...] It is not clear how the proposed method applies to these problems or whether the theoretical results hold in these settings.
>
> The RF-BO framework (Eq. 3) targets general equilibrium-seeking tasks, covering ODE steady-states and minimax games. ODE Control uses $dx = -\text{atanh}(x(t)) + u(t)$ with $h(\alpha,\theta) = \mathbb{E}[x_{ss}(\alpha)] - x_{target} = 0$; WGAN-GP tunes $\lambda$ via $h(\lambda,\theta) = \mathbb{E}[\|\nabla_{\hat{x}} D(\hat{x})\|_2] - 1 = 0$. Sections 6.1.2 and 6.3.1 will add mappings to Eq. 3.
>
> >4. I don't think the two-time scale update in this paper is new per se. For example, the references listed by the authors did something similar (Konda & Tsitsiklis, 1999; Doan, 2022; Hu et al., 2024). So this paper reads more like an application of the classic two-time scale update rule to the bilevel root-finding problem, rather than a novel algorithmic contribution.
>
> Originating in actor-critic algorithms (Konda & Tsitsiklis, 1999), TTSA remains classical (Section 2); the update rule is not claimed as invented here. The core contribution applies the classic TTSA rule to a newly formalized problem class (RF-BO) to structurally cure the Variance Trap, alongside three additions: (1) formalizing RF-BO (Section 3), (2) theoretically diagnosing the Variance Trap (Proposition 5.4), and (3) providing tailored non-asymptotic bounds under Markovian noise (Theorems 5.5–5.7). Introduction phrasing will be refined to accurately position these contributions against existing TTSA literature.
>
> >Minors: Line 307: Define [...] on the left-hand side is confusing here since the ODE dynamic only involves, not.
>
> The variable $x_{ss}(\alpha)$ will be explicitly defined as the steady-state value alongside the ODE equation; LHS notation will be corrected reflecting control parameter $\alpha$ dependency.
>
> **Key Questions For Authors:**
> >Assumption 5.3 requires the upper-level function to have a unique root. How does the method behave when there are multiple roots?
>
> When Assumption 5.3 fails (multiple roots), the method follows the residual flow to the nearest stable attractor, acting as a local search. Appendix C.2 and C.5 show RF-BO escaping local geometry and converging to local equilibria, unlike oscillations of implicit-gradient methods. We will add a brief comment in the final version.
>
> >The variance reduction argument is reasonable in the stochastic setting, but how does TTSA compare to implicit-gradient methods in the deterministic or low-noise regime where the variance issue is less pronounced? Is there a regime where differentiating through the Jacobian is actually preferable?
>
> We appreciate this insightful question. In deterministic or low-noise regimes, differentiating through the Jacobian is preferable. Per Proposition 5.4, the variance trap is driven by noise $\sigma_{\nabla h}^2$; with negligible noise, implicit gradients provide curvature information accelerating convergence. In zero-noise regimes, RF-BO's sole advantage is computational complexity ($\mathcal{O}(C_G)$ vs. $\mathcal{O}(C_G + C_H \times \text{iters})$ for implicit methods). A Discussion paragraph will clarify this trade-off: RF-BO targets high-variance stochastic environments, whereas implicit-gradient methods dominate deterministic, well-conditioned settings permitting cheap curvature exploitation.
>
> We are keen to discuss if there are any further questions.

---

> > ### Author Rebuttal · Reviewer_3bhk · 2026-04-03
> >
> > > Sections 6.1.2 and 6.3.1 will add mappings to Eq. 3.
> >
> > It would be great if the authors could make this explicit in the response period. The issue with the ODE dynamic is that the steady state is not necessarily a minimizer of some function as required in the bilevel formulation. Meanwhile, WGAN-GP itself involves a minimax optimization. Now RF-BO introduces another bilevel formulation for the inner maximization problem in the minimax optimization. So it would be great to make it explicit.
> >
> > **Low noise regimes**
> >
> > I thank the authors for their clarification on the high-noise regimes vs. low-noise regimes. It would be great to add this in the revision to make this message clearer.

---

> > > ### Author Response · Authors · 2026-04-04
> > >
> > > We thank the reviewer for the careful follow-up, and we are glad to have the opportunity to address these two points more carefully than our first-round rebuttal allowed.
> > >
> > > **1. Mapping of ODE Steady-State Control & WGAN-GP to RF-BO Formulation (Eq. 3)**
> > >
> > > We provide the formal mappings for both tasks to the RF-BO canonical formulation (Eq. 3) as defined in Section 3.1, consistent with the experimental setups in Sections 6.1.2 and 6.3.1:
> > >
> > > **1.1 ODE Steady-State Control (Section 6.1.2)**
> > >
> > > The RF-BO formulation (Eq. 3): Find $\alpha^{\star} \in A$ s.t. $h(\alpha^{\star}, \theta^{\star}(\alpha^{\star}))=0$, where $\theta^{\star}(\alpha)=\arg \min_{\theta \in \Theta} R(\alpha, \theta)$.
> > >
> > > - **Lower-level mapping**: The stochastic ODE $dx/dt=-\alpha \tanh(x(t))+u(t)$ defines a steady-state optimization objective $R(\alpha, \theta)=\mathbb{E}[(x(t)-x_{ss}(\alpha))^2]$, where $\theta^{\star}(\alpha)$ is the **steady-state solution** of the ODE (the unique minimizer of $R(\alpha, \theta)$).
> > > - **Upper-level mapping**: The root-finding condition is $h(\alpha, \theta)=\mathbb{E}[x_{ss}(\alpha)]-x_{\mathrm{target}}=0$, which matches the RF-BO root constraint.
> > >
> > > The ODE control task is thus a direct instantiation of Eq. 3, where the ODE steady state corresponds to the lower-level minimizer $\theta^{\star}(\alpha)$.
> > >
> > > **1.2 WGAN-GP Gradient Penalty Tuning (Section 6.3.1)**
> > >
> > > - **Lower-level mapping**: The WGAN-GP discriminator optimization $D^{\star}(\lambda)=\arg \min_{D} \max_{G} \mathcal{L}_{\mathrm{WGAN\text{-}GP}}(G, D; \lambda)$ is a standard minimax lower-level problem, whose optimal solution $\theta^{\star}(\lambda)=D^{\star}(\lambda)$ (discriminator parameters) serves as the lower-level minimizer in RF-BO.
> > > - **Upper-level mapping**: The root-finding constraint for penalty coefficient tuning is $h(\lambda, \theta) = \mathbb{E}[|\nabla_{\hat{x}} D(\hat{x})|^{2}] - 1 = 0$, which fits the upper-level root condition of Eq. 3.
> > >
> > > The minimax structure of WGAN-GP is compatible with RF-BO, as the lower-level optimal solution $\theta^{\star}(\lambda)$ (discriminator equilibrium) is still a valid input to the upper-level root function $h(\cdot)$.
> > >
> > > We will incorporate these mappings into Sections 6.1.2 and 6.3.1 in the revised manuscript.
> > >
> > > **2. Low-Noise/Deterministic Regime Comparison**
> > >
> > > Following the reviewer's suggestion, we will add a paragraph to the Discussion section, grounded in Proposition 5.4:
> > >
> > > > In **deterministic or low-noise regimes**, implicit-gradient methods are preferable: the Variance Trap (Proposition 5.4) vanishes as stochastic noise $\xi \to 0$, and implicit gradients provide curvature information that accelerates convergence. In this regime, RF-BO's advantage reduces to lower computational complexity ($\mathcal{O}(C_G)$ vs. $\mathcal{O}(K \times C_G)$ for implicit methods). In **high-variance stochastic regimes** (the focus of this work), RF-BO avoids variance amplification from implicit Jacobian estimation, outperforming implicit-gradient methods in stability and convergence.
> > >
> > > This paragraph will be included in the camera-ready version to clearly communicate the regime-dependent trade-off.
> > >
> > > These two clarifications will both appear in the camera-ready revision, and we hope they address the reviewer's remaining concerns. We are grateful for the constructive and careful suggestions, which have helped us strengthen the presentation, and we welcome any further questions.

---

### Decision · Program_Chairs · 2026-04-30

**Decision:**

Reject

**Comment:**

The paper presents an interesting perspective by framing some bilevel problems as root-finding rather than outer minimization, and several reviewers found the variance-based motivation useful. However, I am leaning toward a weak reject because the current version does not provide a sufficiently reliable acceptance case for its most central claims.

The main reason is that there are unresolved inconsistencies between the theorem statements, proofs, and how the contributions are summarized, and I am not convinced these can all be treated as minor camera-ready fixes. In addition, the practical significance is weaker than the paper’s framing suggests: the algorithm itself is based on standard TTSA machinery, while the empirical evidence and application positioning do not yet fully support the broader novelty and impact claims.

Overall, the paper has a promising core idea, but in its current form, I do not find the technical and empirical support strong enough for acceptance, considering the ICML bar. Thus, I recommend a Weak Reject.